# Thinking in Scales: Accelerating Gigapixel Pathology Image Analysis via Adaptive Continuous Reasoning

**Jiusong Ge** [1]   **Yingkang Zhan** [1]   **Wenjie Zhao** [1]   **Di Zhang** [1]   **Ke Wang** [2]   **Jiashuai Liu** [1]   **Chunze Yang** [1]
**Chengzu Li** [3]   **Jian Zhang** [1]   **Yuxin Dong** [4]   **Ni Zhang** [1]   **Qidong Liu** [*1]   **Mireia Crispin-Ortuzar** [5]   **Huazhu Fu** [6]
**Chen Li** [*1]   **Zeyu Gao** [*5]

## Abstract

Traditional whole slide image (WSI) analysis methods typically rely on the multiple instance learning (MIL) paradigm, which extracts patch-level features at high magnification and aggregates them for slide-level prediction. However, such exhaustive patch-level processing is computationally expensive, severely limiting the efficiency and scalability of WSI analysis. To address this challenge, we propose **PathCTM** (a **Path**ology-oriented **C**ontinuous **T**hought **M**odel) that enables token-efficient scale-space continuous reasoning for gigapixel WSIs. PathCTM formulates diagnostic inference as a dynamic sequential information pursuit. It progressively transitions from low-magnification global to high-magnification local inspection, and adaptively terminates inference when sufficient evidence is gathered to effectively bound decision uncertainty. Specifically, it uses conditional computation for dynamic scale switching with attention-guided region pruning, coupled with confidence-aware early stopping. Extensive experiments demonstrate that, compared with standard MIL-based methods, PathCTM reduces the number of required image patches by 95.95% and shortens inference time by approximately 95.62%, while maintaining AUC without degradation. Code is available at https://github.com/JSGe-AI/PathCTM.

[1]School of Computer Science and Technology, Xi'an Jiaotong University, Xi'an, China [2]Department of Transmedia Art, Xi'an Academy of Fine Arts, Xi'an, China [3]Language Technology Lab, University of Cambridge, Cambridge, U.K. [4]Moonshot AI [5]Department of Oncology, University of Cambridge, Cambridge, U.K. [6]Institute of High Performance Computing, Agency for Science, Technology and Research, Singapore, Singapore. Correspondence to: Zeyu Gao <zg323@cam.ac.uk>, Qidong Liu <liuqidong@xjtu.edu.cn>, Chen Li <cli@xjtu.edu.cn>.

*Proceedings of the 43rd International Conference on Machine Learning*, Seoul, South Korea. PMLR 306, 2026. Copyright 2026 by the author(s).

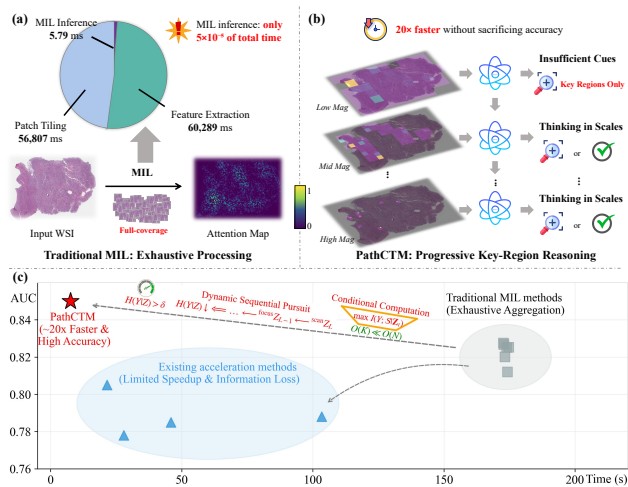

*Figure 1.* (a) Traditional MIL pipeline for WSI analysis, along with the time-cost breakdown of each stage. (b) PathCTM's dynamic multi-scale continuous reasoning process, following a thinking-in-scales paradigm, as the analysis progressively refines from global to local. (c) Performance comparison between PathCTM and existing methods.

## 1. Introduction

Whole Slide Image (WSI) analysis represents one of the most demanding tasks in computational pathology. With advances in digital scanning, pathological slides can now be digitized at gigapixel resolution, enabling visualization of both tissue architecture and cellular morphology within a single specimen. However, such ultra–high-resolution input incurs substantial computational and storage overheads, hindering the scalability and practical deployment of automated WSI analysis in clinical settings.

The prevailing paradigm for WSI analysis (Campanella et al., 2019; Chen et al., 2022; Gao et al., 2023) is based on Multiple Instance Learning (MIL), which divides each WSI into tens of thousands of high-magnification patches for feature extraction and performs one-shot feature aggregation for prediction. While this paradigm has demonstrated strong performance in various downstream tasks, particularly when equipped with pathology foundation models (Vorontsov

et al., 2024; Xiang et al., 2025; Xu et al., 2024a; Zhang et al., 2026), it relies on exhaustive patch-level processing, resulting in substantial computational overhead and inefficient inference. As illustrated in Figure 1 (a), patch tiling and feature extraction dominate the runtime, yet a large proportion of patches contribute negligibly to the final prediction. Therefore, enabling models to adaptively allocate computational resources to diagnostically informative regions across scales is crucial for achieving efficient and clinically scalable WSI analysis.

Although existing studies (Yu et al., 2024; Dong et al., 2025) accelerate WSI analysis, they often rely on fine-grained annotations or rigid cascade structures, merely mimicking the procedural form of pathologists "coarse-to-fine" workflow while lacking continuous, memory-driven reasoning capabilities. As a result, they are limited in scalability and frequently lead to degraded accuracy or marginal efficiency gains. In clinical practice, pathologists conduct multi-scale reasoning in a continuous, progressive, and memory-aware manner. They dynamically integrate cross-scale context and halt inspection once sufficient diagnostic evidence is gathered, thereby ensuring both efficiency and reliability. This discrepancy highlights a **Reasoning Gap** between current methods and the cognitive process of pathologists.

Recently, to bridge the gap between AI systems and human cognition, Continuous Thought Machine (CTM) (Darlow et al., 2026) has been proposed, introducing "internal time" for iterative reasoning. It is promising to equip existing methods with continuous reasoning abilities, providing a theoretical alternative. However, standard CTM is limited to reasoning over single-scale conventional static images and cannot be directly applied to gigapixel-level WSI analysis. Its fundamental premise is that extending thinking time on a fixed tensor enables deeper information extraction. In low-resolution WSI scenarios, no amount of internal temporal iterations can "hallucinate" the missing cellular details at coarse scales. Moreover, this framework is unable to leverage the intrinsic hierarchical structure of WSIs. Overall, existing CTM architecture fails in pathological settings.

To address these challenges, we propose PathCTM, an efficient continuous-reasoning paradigm for gigapixel pathology images. PathCTM retains standard "Thinking in Time" as its internal iterative engine, while further introducing a novel "Thinking in Scales" mechanism that aligns iterative reasoning with the multi-magnification hierarchy of the WSI pyramid. This design formulates diagnosis as sequential evidence acquisition across scales and regions, enabling efficient, interpretable cross-scale inference with a substantially reduced token budget. As illustrated in Fig. 1(b), our framework consists of three core components: (i) **Scale-Space Continuous Reasoning:** PathCTM performs cross-scale continuous reasoning in the spatial dimension, establish-

ing a coherent coarse-to-fine scale-wise inference trajectory. By enforcing state continuity across scale transitions, the model dynamically integrates global context with local fine-grained information; (ii) **Attention-Guided Region Pruning:** This module transforms conventional soft attention into a conditional hard pruning strategy driven by information density. High-resolution features are selectively loaded for informative regions, while error propagation is mitigated through a joint focus filtering mechanism, thereby eliminating the inherent speed bottlenecks of WSI analysis; (iii) **Confidence-Aware Early Stopping:** Considering the varying difficulty of diagnostic cases, PathCTM incorporates a confidence-aware early stopping strategy based on entropy minimization. Inference is allowed to terminate immediately once sufficient evidence is accumulated to theoretically bound decision uncertainty, ensuring reliability with minimal computational cost. Moreover, PathCTM is fully compatible with existing pathology foundation models and can be seamlessly integrated into standard MIL pipelines, improving both inference efficiency and interpretability.

With these designs, PathCTM enables a paradigm shift that, as illustrated in Figure 1 (c), allows it to far exceed existing methods in inference speed without compromising accuracy. In summary, our main contributions are as follows:

- We propose PathCTM, an efficient continuous reasoning framework for WSI, formulated as a dynamic sequential information search process. It emulates the pathologist's coarse-to-fine reasoning workflow, enables interpretable diagnosis, and can be seamlessly integrated with pathology foundation models.

- We design a progressive multi-scale inference strategy driven by conditional computation and entropy minimization. This design enables smooth cross-scale transitions, attention-guided focus selection, and confidence-aware termination, integrating global and local cues while minimizing redundant computation.

- Extensive experiments across four diagnostic tasks show that, compared with standard MIL methods, PathCTM reduces inference time by 95.62% while further improving AUC over the baselines. Analysis and visualization of its inference trajectories further demonstrate PathCTM's enhanced interpretability and its dynamic, thinking-in-scales reasoning behavior.

## 2. Related Work

**Multiple Instance Learning.** Computational pathology has recently advanced beyond conventional diagnosis toward diverse clinically oriented tasks and reliable AI deployment (Zhang et al., 2025; Yang et al., 2026; Ge et al., 2025; Zhu et al., 2025). In particular, WSI analysis has become a

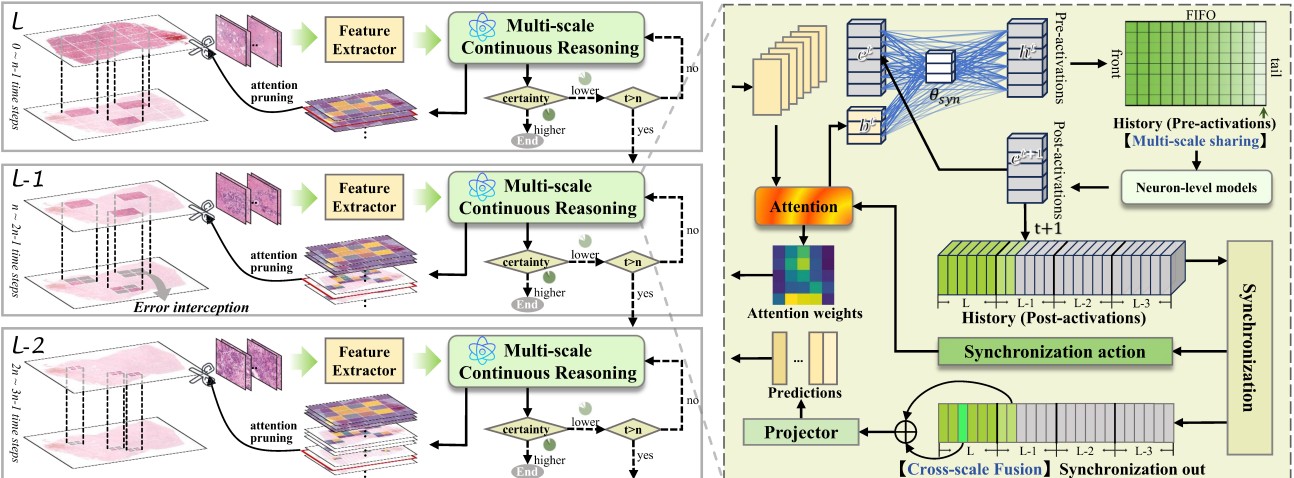

*Figure 2.* **Overall framework of PathCTM.** Formulating WSI analysis as a dynamic sequential information pursuit, the model executes progressive continuous reasoning from low ($L$) to high ($L-1$) magnification. If confidence is insufficient at scale $L$, the model utilizes attention weights from the most confident step to perform Top-K region pruning, guiding the transition to the next scale. Cross-scale consistency is maintained by concatenating synchronization outputs from previous scales. The inference supports confidence-aware early stopping, terminating immediately once the confidence threshold is met to optimize efficiency.

core task in computational pathology, where MIL remains the dominant paradigm. While methods like CLAM (Lu et al., 2021), TransMIL (Shao et al., 2021), and ABMIL (Ilse et al., 2018) have established strong baselines, they suffer from high computational costs due to exhaustive processing. Recent acceleration attempts, such as hierarchical distillation (Dong et al., 2025), and self-reform focusing (Yu et al., 2024) (which relies on fine-grained annotations for training), have accelerated WSI analysis to a certain extent. Similarly, methods such as ZoomMIL (Thandiackal et al., 2022), HAG-MIL (Xiong et al., 2023), and EAGLE (Neidlinger et al., 2025) attempt to reduce redundant computation through multi-scale region selection or key-tile filtering. However, these approaches often fail to address the core bottleneck of feature extraction or suffer from accuracy degradation and poor compatibility with foundation models due to their rigid, procedural inference designs.

**Continuous Thought Machines.** Recently, dynamic reasoning and adaptive state modeling have attracted increasing attention in visual AI systems (Yan et al., 2025; Zhu et al., 2026; Ke et al., 2026). While conventional deep learning models (Krizhevsky et al., 2012; He et al., 2016; Vaswani et al., 2017) rely on static feed-forward computation, biologically plausible paradigms like LTCNs (Hasani et al., 2021) have explored continuous temporal dynamics. Building on these, Continuous Thought Machines (Darlow et al., 2026) further introduce neural synchrony to emulate dynamic thinking processes. Unlike instantaneous response models, CTM processes information through coordinated neural activity over time. However, CTM is designed for single-scale static images and assumes that extending computation on a fixed tensor suffices to extract deeper infor-

mation. When directly applied to WSIs, this assumption breaks down: low-resolution inputs cannot "hallucinate" the missing cellular details at coarse scales, while multi-step reasoning over full-resolution WSIs is computationally intractable. Our work addresses these challenges by developing a scale-aware continuous reasoning framework tailored for WSIs.

## 3. Method

### 3.1. Overview

The core of PathCTM lies in framing WSI analysis as a dynamic inference process designed to minimize diagnostic uncertainty under computational constraints. Unlike static approaches, it establishes an adaptive framework capable of continuous reasoning across scales. As illustrated in Figure 2, the workflow proceeds through four distinct stages: (1) feature encoding, (2) cross-scale continuous reasoning and synchronization, (3) attention-guided region pruning via conditional computation, and (4) confidence-aware early stopping via entropy minimization.

Specifically, PathCTM initializes the reasoning trajectory by extracting low-magnification features from the WSI $X \in \mathbb{R}^{H \times W \times 3}$. It then performs neural dynamic reasoning over $n$ consecutive time steps. At each step $t$, the model evaluates the posterior distribution $P(Y|\mathbf{Z}_t)$ conditioned on the current latent state $\mathbf{Z}_t$, and its associated entropy. If the information gain saturates at the current scale $L$, the model triggers the multi-scale dynamic switching mechanism. To maximize the information-to-compute ratio, the model utilizes the attention distribution from the most confi-

dent time step at scale $L$ as a proxy for information density. It selects $K$ high-value regions (conditional computation) to transition into scale $L-1$ for fine-grained evidence accumulation. This progressive process continues until the diagnostic entropy drops below the acceptance margin or the computational budget is exhausted.

For optimization, PathCTM employs a composite objective that jointly rewards prediction accuracy and certainty. For each internal time step $t \in [1, \ldots, T]$, the model computes the cross-entropy loss $\mathcal{L}^t$ alongside the confidence score $C^t$ (defined as $1 -$ normalized entropy). Across all $z$ scales, for each scale we identify the optimal checkpoints: the minimum-loss point $t_l^1$ and the maximum-certainty point $t_l^2$, defined as:

$$\mathcal{L}_l^t = \text{CrossEntropy}(\hat{y}^t, y^{label}), \quad l \in [1, \ldots, z] \quad (1)$$

$$t_l^1 = \arg\min(\mathcal{L}_l), \quad t_l^2 = \arg\max(C_l) \quad (2)$$

$$\mathcal{L}_{\text{all}} = \frac{1}{z} \sum_{l=1}^{z} \frac{\mathcal{L}_l^{t_l^1} + \mathcal{L}_l^{t_l^2}}{2}. \quad (3)$$

This dual-objective loss ensures that the model not only learns to classify correctly (minimizing $\mathcal{L}$) but also learns to recognize when it is certain (minimizing entropy), aligning the latent dynamics with the stopping criterion.

### 3.2. Continuous Multi-scale Reasoning via Sequential Information Pursuit

**Revisiting WSI Analysis as a Dynamic Process.** Unlike traditional MIL frameworks that approximate the posterior distribution $P(Y|X)$ via a static, one-shot aggregation of high-magnification patches, we formulate WSI analysis as a dynamic sequential information pursuit. The core objective is to progressively minimize the conditional entropy $H(Y|\mathbf{Z}_t)$ of the diagnosis $Y$ across time steps $t$ and multiple scales, given the evolving latent state $\mathbf{Z}_t$. To achieve this, PathCTM constructs a continuous reasoning trajectory that transitions from coarse-grained global context to fine-grained local evidence.

**Latent Space Dynamics.** We model the reasoning process as a recurrent dynamical system. At each scale $L$, the model performs continuous reasoning for $n$ time steps. To ensure the temporal continuity of the information flow, the system maintains a persistent memory state that evolves according to the new sensory input and the previous context. Specifically, let $\mathbf{e}^t$ denote the post-activation state and $\mathbf{h}^t$ denote the pre-activation state at time $t$, where the initial state $\mathbf{e}^1$ is defined as a learnable parameter. The dynamics are governed by a synchronization module $f_{\theta_{\text{syn}}}$, which acts as the state transition function:

$$\mathbf{h}^t = f_{\theta_{\text{syn}}}(\text{concat}(\mathbf{e}^t, \mathbf{b}^t)) \in \mathbb{R}^D, \quad (4)$$

here, $\mathbf{b}^t$ represents the output of the attention mechanism, and $D$ indicates the dimension of the latent space. To capture long-range temporal dependencies without unbounded memory costs, we employ a First-In-First-Out (FIFO) history mechanism. We maintain three historical records: the pre-activation history $\mathbf{H}^t = [\mathbf{h}^{t-M+1}, \ldots, \mathbf{h}^t] \in \mathbb{R}^{D \times M}$, the post-activation history $\mathbf{E}^t = [\mathbf{e}^1, \ldots, \mathbf{e}^t] \in \mathbb{R}^{D \times N}$, and the synchronized output history $\mathbf{S}_{\text{out}}^{\text{list}} = [\mathbf{S}_{\text{out}}^1, \ldots, \mathbf{S}_{\text{out}}^t] \in \mathbb{R}^{D \times N}$. The pre-activation history $\mathbf{H}^t$ stores states from multiple time steps and is updated via FIFO. This ensures that the latent state effectively summarizes the accumulated evidence from the global view down to the current focus.

**Dynamic Scale Switching as Information Maximization.** The transition between scales is driven by insufficient evidence under current resolution constraints. Specifically, when reasoning reaches the maximum time steps $n$ at scale $L$ but prediction confidence fails to reach threshold $\delta$, the system triggers Multi-scale Dynamic Switching. This process queries a new feature subspace at the next higher resolution to maximize conditional mutual information with the diagnostic target. During transitions, the pre-activation history $\mathbf{H}^t$ continues to update following the FIFO rule, preserving temporal continuity across magnifications.

**Information Accumulation via Cross-scale Fusion.** A critical challenge is preventing the "catastrophic forgetting" of global context when the model focuses on local details. To address this, PathCTM adopts a cross-scale fusion strategy that combines the synchronized output $\mathbf{S}_{\text{out}}^{L-1,t}$ from the current fine scale with the most confident output $\mathbf{S}_{\text{out}}^{L,\max}$ from the previous coarse scale:

$$\hat{y}^t = \text{MLP}([\mathbf{S}_{\text{out}}^{L-1,t} \parallel \mathbf{S}_{\text{out}}^{L,\max}]). \quad (5)$$

This fusion mechanism enables the model to continuously leverage the global evidence accumulated from previous reasoning stages during fine-grained inference, rather than relying solely on local information at the current scale. In this way, it effectively mitigates the risk of information loss during focus shifts across regions and scales.

### 3.3. Attention-Guided Region Pruning via Conditional Computation

**Optimization Objective under Budget Constraints.** The primary bottleneck in WSI analysis is the computational intractability of processing all high-magnification patches. We formulate this challenge as a conditional computation problem: given a computational budget $K$ (number of allowable high-resolution patches), we seek to select a subset of regions $\mathcal{S}$ that maximizes the expected information gain for the diagnosis $Y$. Formally, this is an optimization problem:

$$\mathcal{S}^* = \arg \max_{\mathcal{S} \subset \mathcal{X}, |\mathcal{S}| \leq K} I(Y; \mathcal{S}|\mathbf{Z}_t), \quad (6)$$

where $\mathcal{X}$ denotes the set of candidate patches at the current scale, $\mathcal{S}^*$ represents the optimal subset for detailed analysis, and $I(\cdot)$ denotes mutual information. Since computing mutual information for all candidates is infeasible, PathCTM employs an attention-guided region pruning mechanism as a tractable proxy. The attention distribution generated during continuous reasoning approximates local information density, enabling the model to filter redundant, non-informative regions and focus solely on high-value areas.

**Query Generation and Attention Mechanism.** To generate the query vector that guides this selection, PathCTM leverages its latent neural dynamics. At the beginning of training, we sample the $(m, n)$ neurons by randomly selecting $\theta_{\text{out}}$ and $\theta_{\text{action}}$. These are used to generate two synchronized representations at each time step $t$: $\mathbf{S}_{\text{out}}^t \in \mathbb{R}^{\theta_{\text{out}}}$ and $\mathbf{S}_{\text{action}}^t \in \mathbb{R}^{\theta_{\text{action}}}$. These representations are linearly transformed by matrices $\mathbf{W}_o$ and $\mathbf{W}_i$ to produce the model output $\hat{y}^t$ and the attention query vector $\mathbf{q}^t$, respectively:

$$\hat{y}^t = \mathbf{W}_o \cdot \mathbf{S}_{\text{out}}^t, \qquad \mathbf{q}^t = \mathbf{W}_i \cdot \mathbf{S}_{\text{action}}^t. \tag{7}$$

Based on these queries, we employ standard cross-attention to compute the attention weight matrix $\mathbf{A}^t$ and the output representation $\mathbf{b}^t$:

$$\mathbf{A}^t, \mathbf{b}^t = \text{Attn}(\mathbf{q}^t, \text{Encoder}(\text{data})). \tag{8}$$

The resulting $\mathbf{b}^t$ is concatenated with the synaptic state $\mathbf{e}^{t+1}$ and used as part of the input for the next time step. Meanwhile, each attention matrix $\mathbf{A}^t$ is stored in a list $\mathbf{A}_{\text{list}}$ to record the attention distributions across time steps.

**Top-K Selection as Sparse Masking.** To address the optimization problem during scale transitions, we employ an attention-guided greedy selection. Departing from uniform sampling, we enforce spatial sparsity **dictated by** attention weights. We formalize the validity of this attention-based selection criterion through the following proposition:

**Proposition 1 (Attention as Influence Surrogate).** *Under bounded feature norms and additive aggregation, attention weights provide a first-order surrogate for the influence of individual patches on the decision loss. In particular, selecting Top-K patches according to attention scores yields a tractable approximation to gradient-based influence maximization. (Detailed derivation provided in Appendix A.1.)*

Guided by this proposition, we identify the time step $t^*$ that yields the highest prediction confidence within the current reasoning trajectory. Leveraging the theoretical surrogate relationship established above, we select the indices of the top $K$ patches to form the subset $\mathcal{S}$:

$$\mathcal{S} = \{i \mid \mathbf{A}_i^{t^*} \in \text{Top-K}(\mathbf{A}^{t^*})\}. \tag{9}$$

Subsequently, high-resolution feature extraction is performed exclusively for the indices in $\mathcal{S}$ at scale $L - 1$.

This attention-driven strategy effectively acts as a sparse mask, reducing the computational complexity from $\mathcal{O}(N)$ to $\mathcal{O}(K)$ (where $K \ll N$). By concentrating FLOPs on the information-dense tail of the distribution, PathCTM maximizes the information-to-compute ratio, achieving efficient layer-wise focusing while preserving diagnostic accuracy.

**Joint Attention Calibration for Suppressing Error Propagation.** The ambiguity of coarse-grained features often leads to the generation of false positive candidates and the accumulation of errors. To address this, PathCTM introduces a joint attention mechanism during scale transitions. Unlike rigid focusing methods that enforce sub-region selection for all candidates (including erroneous ones), our method jointly models all regions at each scale and recalibrates attention. For the selected subset $\mathcal{S}$, after extracting the corresponding fine-grained features $\mathbf{F}_{\mathcal{S}}^{L-1} = \{\mathbf{f}_k^{L-1}\}_{k \in \mathcal{S}}$ at scale $L - 1$, we compute the new weight distribution $\mathbf{A}^{L-1}$ via joint attention:

$$\mathbf{A}_i^{L-1} = \frac{\exp(\mathbf{q}^t \cdot \mathbf{f}_i^{L-1}/\sqrt{D})}{\sum_{k \in \mathcal{S}} \exp(\mathbf{q}^t \cdot \mathbf{f}_k^{L-1}/\sqrt{D})}, \quad \forall i \in \mathcal{S} \tag{10}$$

This ensures the effective elimination of invalid candidates without triggering subsequent sub-region selection (as shown by the "error interception" at $L - 1$ in Figure 2). It continuously filters out noisy errors, propagating only valid regions to the high-resolution analysis stage.

### 3.4. Confidence-Aware Early Stopping via Entropy Minimization

**Adaptive Stopping Formulation.** Different diagnostic tasks and instances exhibit varying inherent complexity. To adaptively decouple computational cost from instance difficulty, PathCTM introduces a confidence-aware early stopping mechanism. We formulate the inference duration as a stochastic stopping time $\tau$, aiming to halt computation once the model achieves sufficient certainty. Let $\hat{y}_{t,L}$ denote the prediction at time step $t$ under the scale $L$. We quantify the prediction certainty using the confidence score $C_{t,L}$, defined as the complement of the normalized entropy:

$$C_{t,L} = 1 - \frac{H(\hat{y}_{t,L})}{\log N}, \tag{11}$$

where $H(\cdot)$ is the Shannon entropy and $N$ is the number of classes. The stopping time $\tau$ is defined as the first moment when this confidence score exceeds the threshold $\delta$:

$$\tau = \min\{(t, L) \mid C_{t,L} \geq \delta\}. \tag{12}$$

To relate diagnostic uncertainty to the lower bound of classification error, we invoke Fano's inequality as detailed in the following proposition:

**Proposition 2 (Error Bound via Fano's Inequality).** *Fano's inequality establishes an information-theoretic relationship between the conditional entropy $H(Y \mid Z_t)$ and*

*Table 1.* Summary of experimental results across four diagnostic tasks. The best results are highlighted in bold. AUC scores are presented in percentages (%), and Time is measured in seconds (s). FE denotes the feature extractor.

| FE | MODEL | BRACS-3 | | | BRACS-7 | | | RCC-STAGING | | | MUT-SETD2 | | |
|---|---|---|---|---|---|---|---|---|---|---|---|---|---|
| | | AUC ↑ | No. Pat. ↓ | Time ↓ | AUC ↑ | No. Pat. ↓ | Time ↓ | AUC ↑ | No. Pat. ↓ | Time ↓ | AUC ↑ | No. Pat. ↓ | Time ↓ |
| CONCH(V1.5) | CLAM | 92.0±1.3 | 2606 | 74.43 | 86.2±2.1 | 2605 | 74.41 | 77.3±6.0 | 3517 | 100.43 | 73.2±3.0 | 2869 | 81.92 |
| | ABMIL | 91.9±1.3 | 2606 | 74.43 | 88.6±1.3 | 2605 | 74.41 | 74.4±8.0 | 3517 | 100.43 | **74.2±3.9** | 2869 | 81.92 |
| | TRANSMIL | 90.6±0.9 | 2606 | 74.44 | 86.1±1.1 | 2605 | 74.41 | 76.9±5.8 | 3517 | 100.44 | 70.1±4.1 | 2869 | 81.93 |
| | WIKG | 90.4±0.9 | 2606 | 74.44 | 84.4±1.5 | 2605 | 74.41 | 81.2±7.3 | 3517 | 100.44 | 71.1±2.0 | 2869 | 81.92 |
| | DSMIL | 90.6±1.2 | 2606 | 74.43 | 85.3±1.5 | 2605 | 74.41 | 80.0±8.7 | 3517 | 100.43 | 69.9±3.9 | 2869 | 81.92 |
| | RRT | 91.6±1.5 | 2606 | 74.44 | 86.0±1.5 | 2605 | 74.41 | 76.3±6.5 | 3517 | 100.43 | 73.5±4.1 | 2869 | 81.92 |
| | HDMIL | 90.2±3.4 | 1164 | 33.26 | 76.9±5.4 | 882 | 25.21 | 73.8±2.2 | 1256 | 35.88 | 68.9±2.1 | 2087 | 59.52 |
| | HAG-MIL | 85.2±1.1 | 875 | 25.78 | 75.7±3.6 | 895 | 26.37 | 81.3±7.3 | 992 | 29.23 | 64.7±5.2 | 907 | 26.72 |
| | ZOOMMIL | 92.2±1.8 | 395 | 11.64 | 85.5±3.0 | 400 | 11.79 | 78.5±4.8 | 455 | 13.41 | 68.9±3.9 | 387 | 11.40 |
| | EAGLE | 85.9±3.3 | 756 | 22.28 | 82.5±1.6 | 756 | 22.28 | 71.4±10.9 | 774 | 22.81 | 70.5±3.4 | 1465 | 43.16 |
| | PATHCTM | **93.1±1.6** | **183** | **5.34** | **88.9±1.7** | **224** | **6.64** | **82.8±2.6** | **250** | **7.39** | 73.9±2.6 | **235** | **6.93** |
| UNI(V2) | CLAM | 92.0±1.2 | 9484 | 210.15 | 87.0±1.6 | 11929 | 264.33 | 79.0±9.0 | 15063 | 333.80 | 74.1±2.3 | 11229 | 248.84 |
| | ABMIL | 91.8±2.2 | 9484 | 210.17 | 86.9±1.5 | 11929 | 264.33 | 79.7±6.6 | 15063 | 333.81 | 73.7±3.0 | 11229 | 248.85 |
| | TRANSMIL | 89.1±0.9 | 9484 | 210.15 | 85.3±1.2 | 11929 | 264.34 | 79.1±5.2 | 15063 | 333.84 | 71.6±1.7 | 11229 | 248.87 |
| | WIKG | 91.4±1.7 | 9484 | 210.18 | 87.1±1.4 | 11929 | 264.36 | 81.2±7.5 | 15063 | 333.90 | 73.8±1.9 | 11229 | 248.90 |
| | DSMIL | 91.3±0.5 | 9484 | 210.15 | 86.9±1.0 | 11929 | 264.33 | 82.7±5.3 | 15063 | 333.81 | 69.7±3.0 | 11229 | 248.85 |
| | RRT | 91.6±2.4 | 9484 | 210.16 | 87.9±0.5 | 11929 | 264.34 | 80.3±6.6 | 15063 | 333.83 | 74.7±2.8 | 11229 | 248.86 |
| | HDMIL | 90.1±2.4 | 7242 | 160.50 | 82.1±4.7 | 3632 | 80.51 | 79.8±5.1 | 10709 | 237.33 | 68.8±3.1 | 8806 | 195.16 |
| | HAG-MIL | 86.7±1.0 | 1486 | 34.32 | 75.7±3.6 | 895 | 20.67 | 80.2±3.9 | 1752 | 40.46 | 68.8±2.3 | 2236 | 51.64 |
| | ZOOMMIL | 91.7±1.5 | 912 | 21.06 | 86.0±1.5 | 911 | 21.04 | 79.5±4.0 | 1143 | 26.40 | 73.6±4.6 | 1203 | 27.78 |
| | EAGLE | 85.5±1.6 | 935 | 21.59 | 80.3±1.9 | 935 | 21.59 | 78.0±4.5 | 900 | 20.79 | 68.8±2.3 | 2123 | 49.03 |
| | PATHCTM | **93.6±1.1** | **349** | **7.95** | **89.3±1.7** | **362** | **8.28** | **83.0±2.6** | **408** | **9.33** | **75.4±2.8** | **392** | **9.00** |

*the classification error probability $P_e$. In particular, high conditional entropy necessarily implies a non-negligible lower bound on the achievable error probability, thereby providing a theoretical motivation for confidence-aware early stopping. (Derivation provided in Appendix A.2.)*

By enforcing a lower bound on the potential classification error, this mechanism transforms PathCTM into an adaptive inference framework. It aligns the inference termination criterion with the reliability objective, enabling the model to output reliable predictions during dynamic reasoning.

**Dynamic Reasoning and Efficiency Trade-off.** The inference proceeds as a sequential hypothesis testing process. If at any step $t$ within scale $L$, the confidence condition $C_{t,L} \geq \delta$ is met, the system interprets the latent state $\mathbf{Z}_t$ as capturing sufficient evidence and terminates to output $\hat{y}_{t,L}$. Otherwise, the model switches to scale $L - 1$ to resolve ambiguity. This process iterates until the stopping condition is met or the maximum reasoning budget is exhausted. Through this adaptive strategy, PathCTM optimizes a joint objective of minimizing prediction error and computational FLOPs, **effectively balancing accuracy and efficiency by** allocating minimal resources to "easy" instances while reserving deep, multi-scale computation for "hard" cases.

## 4. Experiments

### 4.1. Datasets and Implementation Details

We evaluated the proposed PathCTM framework on four diagnostic tasks across three publicly available datasets: (1) the BRACS (Brancati et al., 2022) dataset, used for breast cancer subtype classification, which includes a high-level category classification task (BRACS-3) and a fine-grained subtype analysis task (BRACS-7); (2) the MUT-HET-RCC (Acosta et al., 2022) dataset, used for gene mutation prediction (MUT-SETD2); and (3) the RCC dataset from TCGA project (Weinstein et al., 2013), used for renal cell carcinoma staging (RCC-staging). Detailed dataset information and training configurations are provided in the Appendix.

To validate the effectiveness of our method, we systematically compared PathCTM with several state-of-the-art (SOTA) MIL methods, including CLAM (Lu et al., 2021), ABMIL (Ilse et al., 2018), TransMIL (Shao et al., 2021), WIKG (Li et al., 2024), DSMIL (Li et al., 2021), RRT (Tang et al., 2024), HDMIL (Dong et al., 2025), ZoomMIL (Thandiackal et al., 2022), HAG-MIL (Xiong et al., 2023), and EAGLE (Neidlinger et al., 2025). *Notably, SMT (Yu et al., 2024) was excluded as it requires fine-grained annotations for training, precluding a fair comparison on these datasets.* To ensure a fair comparison, all methods were evaluated using two feature extractors: CONCH (V1.5)(Lu et al., 2024) and UNI(V2)(Chen et al., 2024). For specific details, please refer to Appendix C.

Performance was evaluated from three perspectives: (1) macro AUC score; (2) the average number of processed patches per slide (No. Pat.); and (3) the total inference time (Time), which includes patch tiling, feature extraction, and model inference. It should be noted that, in this study, "Patch Tiling" refers to the full data preparation pipeline from the raw WSI to the point just before patches are fed into the encoder. Beyond coordinate generation, this process also includes computationally heavy sliding-window decoding,

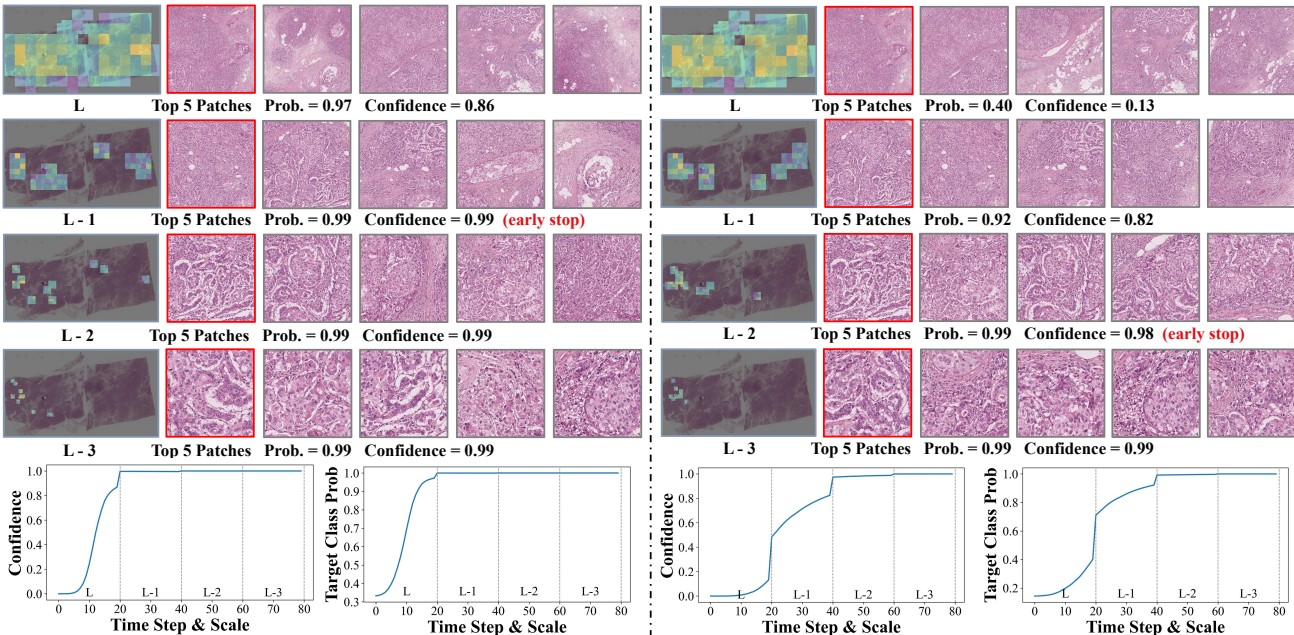

*Figure 3.* Visualization of PathCTM's adaptive continuous reasoning on a single BRACS slide under varying task difficulties (left: BRACS-3; right: BRACS-7). The top panels display the top-5 attended patches at each scale, focusing on key regions exhibiting architectural distortion and nuclear atypia. The bottom panels track the dynamic trajectory of confidence and class prediction probabilities.

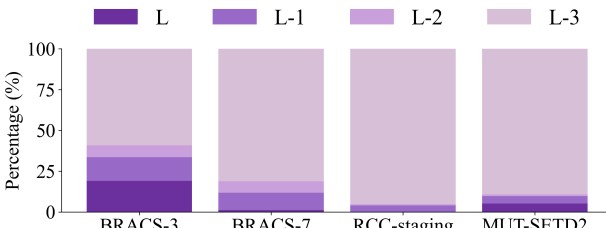

*Figure 4.* The distribution of early-stopping scales of PathCTM across four datasets. ($\delta = 0.9$)

multi-scale coordinate mapping, and I/O-intensive image patch extraction. In frameworks such as CLAM, these I/O costs are often hidden inside the feature extraction loop, whereas we explicitly separate I/O from model inference to more accurately expose the true end-to-end bottlenecks. A detailed description is provided in Supplementary Figure 8.

### 4.2. Performance and Efficiency Analysis

Table 1 presents a comprehensive comparison with existing SOTA methods. PathCTM outperforms baselines in 7 out of 8 experimental settings, securing an average AUC improvement of 2.3%, while only slightly trailing ABMIL on the MUT dataset (CONCH). Most notably, PathCTM delivers an order-of-magnitude boost in efficiency, reducing inference time by 95.62% compared to standard MIL methods and by 92.46% against acceleration approaches. These efficiency gains are most significant with the UNI extractor.

Additionally, we compare PathCTM with existing slide-level foundation models, including CHIEF (Wang et al., 2024), GigaPath (Xu et al., 2024b), PRISM (Shaikovski et al., 2024), and TITAN (Ding et al., 2025). Table 2 reports the results on the BRACS dataset, where PathCTM (using CONCH V1.5) is evaluated against these slide-level models. Similar to MIL-based methods, current slide-level foundation models also face notable speed bottlenecks, whereas PathCTM exhibits a substantial advantage in efficiency.

Furthermore, PathCTM adapts its computational budget to task difficulty via its confidence-aware early-stopping. On easier tasks such as BRACS-3, many cases can already achieve high confidence at lower magnifications, leading to a much smaller number of required patches (183 patches on CONCH). In contrast, more fine-grained tasks such as BRACS-7 require slightly deeper reasoning (224 patches) to discriminate subtle subtype differences. As shown in Figure 4, a similar trend appears in staging and mutation prediction tasks, where the majority of cases must progress to the highest magnification to achieve sufficient diagnostic certainty. This aligns with clinical characteristics: staging demands examination of tumor–stroma interfaces and invasion fronts, while mutation prediction depends on cell-level morphological cues. In comparison, subtype classification relies more heavily on broader tissue-architecture patterns and often converges earlier. Overall, these results show that PathCTM adjusts its inference depth in a thinking-in-scales manner, stopping early on easy cases and reasoning deeper on difficult ones, striking an efficient balance between com-

*Table 2.* Comparison results of PathCTM with slide-level foundational models, with AUC presented in %.

| MODEL | BRACS-3 | | BRACS-7 | | RCC-STAGING | | MUT-SETD2 | |
| --- | --- | --- | --- | --- | --- | --- | --- | --- |
| | AUC ↑ | TIME ↓ | AUC ↑ | TIME ↓ | AUC ↑ | TIME ↓ | AUC ↑ | TIME ↓ |
| CHIEF | 88.13±2.71 | 20.18 | 85.17±1.59 | 20.11 | 76.44±5.39 | 27.23 | 72.13±3.46 | 22.21 |
| GIGAPATH | 85.08±2.47 | 175.79 | 80.86±2.30 | 221.11 | 73.39±6.16 | 279.19 | 72.49±2.65 | 208.13 |
| PRISM | 88.96±2.38 | 113.85 | 85.58±1.41 | 115.52 | 74.38±5.00 | 144.32 | 72.20±2.83 | 107.58 |
| TITAN | 91.82±1.42 | 74.45 | 86.72±1.18 | 74.42 | 79.08±5.57 | 100.45 | 73.44±3.46 | 81.93 |
| **PATHCTM** | **93.10±1.58** | **5.34** | **88.90±1.74** | **6.64** | **82.84±2.55** | **7.39** | **73.94±2.61** | **6.93** |

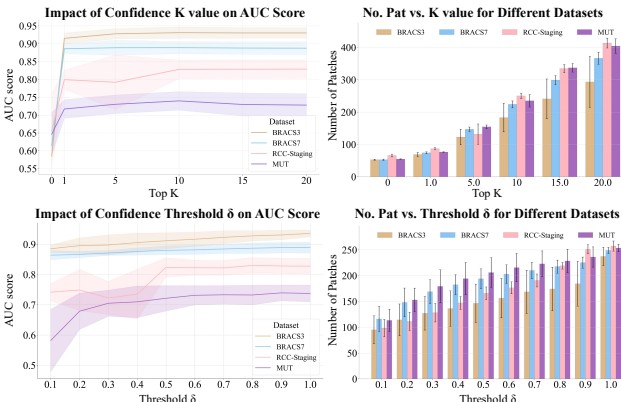

*Figure 5.* The impact of the selected patch number $K$ and confidence threshold $\delta$ on prediction accuracy and inference efficiency.

putational cost and diagnostic accuracy.

### 4.3. Visualization and Interpretability Analysis

Figure 3 visualizes PathCTM's adaptive inference trajectory on a randomly selected BRACS slide under diagnostic tasks of varying difficulty. Morphologically, as the scale advances and the model acquires increasingly rich semantic information, both confidence and prediction probability exhibit an upward trend. At each scale, the top-5 patches are precisely localized within epithelial regions and ductal structures. These regions effectively capture architectural distortion and nuclear atypia, which are critical for differentiating fine-grained breast cancer subtypes.

Behaviorally, PathCTM demonstrates an awareness of task complexity, a capability quantitatively validated by the dynamics curves. In the simpler lesion-type classification (BRACS-3) task (left), the model determines that coarse-grained features are sufficient. This judgment manifests in the curve as a steep upward trend—confidence rapidly saturates and triggers the threshold at the Scale L-1, terminating inference early to avoid redundancy. Conversely, in the more challenging lesion-subtype (BRACS-7) task (right), the model "realizes" the insufficiency of current evidence and advances reasoning to a deeper level (L-2) to capture subtler cellular-level evidence and resolve uncertainty. The curve displays a step-wise growth, particularly characterized by significant jumps in confidence across scale boundaries,

*Table 3.* Ablation study of PathCTM components on BRACS-3 using CONCH. I: Attention-guided Region Pruning; II: Multi-scale Synchronous Fusion; III: Confidence-aware Early Stopping.

| I | II | III | NO. PAT. ↓ | ACC ↑ | AUC ↑ |
| --- | --- | --- | --- | --- | --- |
| | | | 3529 | $78.32_{1.03}$ | $93.72_{0.44}$ |
| ✓ | | | 246 | $76.28_{1.41}$ | $93.07_{0.61}$ |
| ✓ | ✓ | | 246 | $78.31_{1.48}$ | $93.59_{1.14}$ |
| ✓ | ✓ | ✓ | 183 | $78.31_{1.48}$ | $93.10_{1.58}$ |

proving that fine-grained features are indispensable for eliminating uncertainty in difficult cases. This dynamic process closely aligns with the "coarse-to-fine" diagnostic logic of pathologists: allocating more attention to complex cases while maintaining efficiency on typical ones. This not only validates the efficiency of PathCTM but also demonstrates the high clinical interpretability of its decision-making process. Supplementary visualizations for RCC staging are in the Appendix to illustrate the model's adaptability to cases of varying difficulty within the same task.

### 4.4. Ablation Study and Parameter Analysis

We evaluated the contributions of PathCTM's core modules on BRACS-3 (Table 3). The baseline suffered from high computational costs. Introducing Attention-guided Region Pruning drastically reduced patch usage with minimal accuracy trade-off. Subsequent Multi-scale Synchronous Fusion recovered performance by preserving cross-scale context. Finally, Confidence-aware Early Stopping further optimized efficiency by terminating inference at high certainty, maintaining accuracy with minimal computation.

PathCTM allows for flexible balancing of inference efficiency and accuracy according to specific diagnostic requirements. Further parameter analysis (Figure 5) reveals the impact of patch count ($K$) and confidence threshold ($\delta$) on this trade-off. Regarding $K$, while increasing the patch count initially enhances accuracy, gains saturate beyond $K = 10$ due to information redundancy; hence, we adopt $K = 10$ as the default. Similarly, prediction stability is achieved within $\delta \in [0.8, 1.0]$. We recommend setting $\delta = 0.9$, as extending reasoning beyond this point yields negligible accuracy improvements, whereas terminating at

this threshold effectively optimizes inference speed while ensuring reliable predictions. See Appendix for details.

## 5. Conclusion

In this work, we present **PathCTM**, an efficient continuous reasoning framework for WSI compatible with pathology foundation models. By mirroring the pathologist's coarse-to-fine workflow via adaptive pruning and confidence-aware stopping, PathCTM dynamically highlights diagnostic regions during cross-scale reasoning and substantially accelerates inference. Experiments show that PathCTM achieves high diagnostic accuracy and significant computational efficiency across multiple datasets, while exhibiting a clear global-to-local diagnostic logic that enhances interpretability. This paradigm offers a promising direction for scalable, efficient, and clinically aligned WSI analysis.

## Acknowledgements

This work was supported by the Shaanxi Province Key R&D Program (2025SF-YBXM-363, 2024SF-GJHX-32), the National Science and Technology Major Project (2025ZD0544802, 2024ZD0527700), the Key Research and Development Program of Ningxia Hui Autonomous Region (2023BEG02023), the "Research on Key Technologies for Full-Chain Intelligent Pathological Diagnosis" project of The First Affiliated Hospital of Xi'an Jiaotong University (HX202440), the Post-Doctoral Research Project of Shaanxi Province (31271000000008), the National Natural Science Foundation of China (62506291), the XJTU Research Fund for AI Science (2025YXYC004), the Fundamental Research Funds for the Central Universities (No. xzy012026032), the Royal Society (RGS\R2\252688), and GE HealthCare.

## Impact Statement

This work contributes to more efficient and scalable computational pathology by reducing redundant computation in gigapixel whole slide image analysis. PathCTM can lower inference cost and latency, making computational pathology models more suitable for clinical and research applications. However, this method is intended to assist pathologists rather than replace them, and further validation on diverse clinical datasets is required before deployment.

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

# A. Theoretical Analysis and Proofs

In this section, we provide the formal proofs and theoretical derivations for the core mechanisms of PathCTM. Specifically, we detail the mathematical grounding for: (1) Attention-Guided Region Pruning via first-order sensitivity analysis; and (2) Confidence-Aware Early Stopping via error probability bounding.

### A.1. Proof of Proposition 1 (Attention-Guided Region Pruning)

**Restatement of Proposition 1.** *Under bounded feature norms and additive aggregation, attention weights provide a first-order surrogate for the influence of individual patches on the decision loss. In particular, selecting Top-$K$ patches according to attention scores yields a tractable approximation to gradient-based influence maximization.*

**Derivation.** We formalize patch selection as identifying a subset of patches whose removal induces the largest first-order variation in the loss function. Let the aggregated slide-level representation be defined as

$$z_{\text{agg}} = \sum_{j=1}^{N} A_j h_j, \tag{13}$$

where $A_j$ denotes the attention weight associated with patch $j$, and $h_j$ its corresponding feature embedding. We define the *influence* of a patch $x_i$ as the variation of the loss functional $\mathcal{L}$ under the perturbation $A_i \to 0$.

**Step 1: First-Order Sensitivity Analysis.** We treat the loss $\mathcal{L}(\mathbf{A})$ as a differentiable function of the attention vector $\mathbf{A} \in \mathbb{R}^N$. To estimate the contribution of patch $i$, we analyze the variation in loss induced by **masking its attention weight** (i.e., setting $A_i \to 0$) while holding other weights constant. This corresponds to the perturbation $\mathbf{A}' = \mathbf{A} - A_i \mathbf{e}_i$, where $\mathbf{e}_i$ is the $i$-th canonical basis vector.

*Remark: For efficient first-order approximation, we treat the attention weights as independent variables during this sensitivity analysis. We do not re-normalize the remaining weights, as strictly enforcing the simplex constraint ($\sum A = 1$) would introduce high-order coupling effects via the softmax Jacobian, obscuring the direct contribution of individual patches.*

Applying a first-order Taylor expansion around $\mathbf{A}$ yields:

$$\mathcal{L}(\mathbf{A} - A_i \mathbf{e}_i) \approx \mathcal{L}(\mathbf{A}) - A_i \frac{\partial \mathcal{L}}{\partial A_i}. \tag{14}$$

The resulting first-order influence is therefore given by:

$$\Delta \mathcal{L}_i \approx -A_i \frac{\partial \mathcal{L}}{\partial A_i}. \tag{15}$$

**Step 2: Gradient Decomposition via the Aggregation Layer.** Assuming additive aggregation, the gradient with respect to the attention coefficient can be expressed using the chain rule as:

$$\frac{\partial \mathcal{L}}{\partial A_i} = \left\langle \frac{\partial \mathcal{L}}{\partial z_{\text{agg}}}, \frac{\partial z_{\text{agg}}}{\partial A_i} \right\rangle = \nabla_z \mathcal{L}^\top h_i. \tag{16}$$

Substituting this expression into the influence estimate yields:

$$|\Delta \mathcal{L}_i| \approx A_i \left| \nabla_z \mathcal{L}^\top h_i \right|. \tag{17}$$

**Step 3: Bounded Influence and Surrogate Justification.** We assume that the feature embeddings are norm-bounded, i.e., $\|h_i\|_2 \leq C_h$, which is typically enforced through normalization layers (e.g., LayerNorm). Furthermore, we assume the loss landscape satisfies local Lipschitz continuity, implying a bounded gradient norm $\|\nabla_z \mathcal{L}\|_2 \leq C_g$. Applying the Cauchy–Schwarz inequality gives:

$$|\Delta \mathcal{L}_i| \leq A_i \|\nabla_z \mathcal{L}\|_2 \|h_i\|_2 \leq A_i \cdot (C_g C_h). \tag{18}$$

Under these regularity conditions, the attention coefficient $A_i$ provides an effective upper bound on the first-order influence of patch $i$. Consequently, selecting patches with the largest attention weights prioritizes those with the highest potential impact on the decision boundary.

**Discussion.** We emphasize that the above result does not establish strict rank equivalence between attention scores and gradient-based influence (due to the potential orthogonality between the gradient and feature vectors). Instead, attention-guided Top-$K$ selection serves as a computationally efficient surrogate for first-order sensitivity analysis. This approximation is particularly attractive in large-scale whole-slide image settings, where explicit gradient-based evaluation of all patches is computationally prohibitive.

### A.2. Proof of Proposition 2 (Confidence-Aware Early Stopping)

**Restatement of Proposition 2.** *Fano's inequality establishes an information-theoretic relationship between the conditional entropy $H(Y \mid Z_t)$ and the classification error probability $P_e$. In particular, high conditional entropy necessarily implies a non-negligible lower bound on the achievable error probability, thereby providing a theoretical motivation for confidence-aware early stopping.*

**Derivation.** We recall the derivation of Fano's inequality in the context of multi-class classification. Let $Y \in \{1, \ldots, N\}$ denote the ground-truth label, $\hat{Y}$ the predicted label, and define the error indicator variable $E = \mathbb{I}(\hat{Y} \neq Y)$.

**Step 1: Entropy Decomposition.** Using the chain rule, the joint entropy $H(E, Y \mid \hat{Y})$ can be expanded in two ways:

$$H(E, Y \mid \hat{Y}) = H(Y \mid \hat{Y}) + H(E \mid Y, \hat{Y}) = H(E \mid \hat{Y}) + H(Y \mid E, \hat{Y}). \tag{19}$$

Since $E$ is deterministically determined by $Y$ and $\hat{Y}$, $H(E \mid Y, \hat{Y}) = 0$. Thus:

$$H(Y \mid \hat{Y}) = H(E \mid \hat{Y}) + H(Y \mid E, \hat{Y}). \tag{20}$$

**Step 2: Bounding the Conditional Terms.** First, conditioning reduces entropy, so $H(E \mid \hat{Y}) \leq H(E) = H(P_e)$, where $H(P_e)$ is the binary entropy function ($H(P_e) \leq 1$). Second, we decompose the term $H(Y \mid E, \hat{Y})$:

$$H(Y \mid E, \hat{Y}) = P(E = 0) H(Y \mid \hat{Y}, E = 0) + P(E = 1) H(Y \mid \hat{Y}, E = 1). \tag{21}$$

If $E = 0$ (correct prediction), $Y = \hat{Y}$ deterministically, so the entropy is 0. If $E = 1$ (error), $Y$ can be any of the remaining $N - 1$ classes. By the maximum entropy principle,

$$H(Y \mid \hat{Y}, E = 1) \leq \log(N - 1). \tag{22}$$

**Step 3: Fano's Inequality.** Combining these bounds yields the classical inequality:

$$H(Y \mid \hat{Y}) \leq H(P_e) + P_e \log(N - 1). \tag{23}$$

This inequality implies that the error probability $P_e$ is lower-bounded by the conditional entropy. A simplified lower bound (using $H(P_e) \leq 1$) is given by:

$$P_e \geq \frac{H(Y \mid \hat{Y}) - 1}{\log(N - 1)}. \tag{24}$$

**Connection to Latent Representations.** In PathCTM, predictions are derived from a latent state $Z_t$. By the Data Processing Inequality, the latent state contains at least as much information about $Y$ as the final discrete prediction $\hat{Y}$, implying

$$H(Y \mid Z_t) \leq H(Y \mid \hat{Y}). \tag{25}$$

Substituting this relation into Fano's inequality shows that high uncertainty in the latent representation necessarily induces a non-negligible lower bound on the achievable classification error.

**Implication for Early Stopping.** The analysis above assumes access to the true conditional entropy. In practice, we assume the model is reasonably calibrated, such that the entropy of the predictive distribution $H(\hat{y}_{\text{softmax}})$ serves as a proxy for the posterior uncertainty $H(Y \mid Z_t)$. Under this calibration assumption, the stopping criterion

$$1 - \frac{H(\hat{y}_{\text{softmax}})}{\log N} \geq \delta \tag{26}$$

provides an information-theoretically motivated heuristic for terminating inference under sufficiently low predictive uncertainty, rather than a strict guarantee on the resulting error probability.

## B. Description of datasets and tasks

We conducted a comprehensive evaluation of the proposed method on four diagnostic tasks across three publicly available pathology datasets. Details are provided as follows:

**BRACS Dataset.** BRACS is a large-scale, multi-center public dataset for breast cancer subtype classification, consisting of 547 whole-slide images (WSIs) and 4,539 region-of-interest (ROI) patches. It includes two classification tasks with different granularity levels: a three-class task (BRACS-3) and a more fine-grained seven-class task (BRACS-7). The three-class task provides a fundamental basis for rapid malignancy risk assessment, while the seven-class task further refines the categorization into seven specific subtypes, requiring the model to distinguish subtle morphological differences and therefore posing a more challenging setting. We evaluate PathCTM on both tasks to assess its performance under different levels of diagnostic complexity. The classification categories are shown in Table 4.

**MUT-HET-RCC Dataset.** The MUT-HET-RCC dataset was designed for studying molecular mutation heterogeneity in clear cell renal cell carcinoma (ccRCC). It contains 1,291 FFPE-derived H&E-stained WSIs and provides binary mutation labels for several key driver genes, including *BAP1*, *PBRM1*, and *SETD2*. In our experiments, we evaluate all methods on the *SETD2* mutation prediction task.

**RCC-Staging Dataset.** For the kidney cancer staging task, two senior pathologists re-annotated the RCC dataset by categorizing pT1 and pT2 cases as early-stage, and pT3 and pT4 cases as late-stage, while removing normal and heavily contaminated slides. A total of 376 cases were retained, including 284 early-stage and 92 late-stage cases. This task focuses on assessing the ability of each model to identify tumor progression severity.

*Table 4.* Category distribution of the BRACS dataset.

| Lesion type | Lesion subtype | Percentage |
|---|---|---|
| Benign | Normal | 8% |
| | Pathological Benign | 27% |
| | Usual Ductal Hyperplasia | 14% |
| Atypical | Flat Epithelial Atypia | 7% |
| | Atypical Ductal Hyperplasia | 9% |
| Malignant | Ductal Carcinoma in Situ | 11% |
| | Invasive Carcinoma | 24% |

## C. Detailed Experimental Settings

In this section, we delineate the specific architectural specifications and hyperparameter configurations employed for PathCTM. A comprehensive summary of these settings is provided in Tables 5 and 6, respectively. Experiments were conducted on a workstation with $1 \times$ NVIDIA GeForce RTX 4090 GPU.

**Feature Extraction and Preprocessing.** For feature extraction, we strictly adhered to the official configurations of the employed backbone models to ensure optimal representation quality. Specifically, the input patch size was set to $1024 \times 1024$ for CONCH and $512 \times 512$ for UNI. Following the standard protocol established by CLAM, we applied a consistent data preprocessing pipeline across all baseline methods. To empower PathCTM with continuous multi-scale reasoning capabilities, we extracted image patches from four distinct magnification levels: $40\times$, $20\times$, $10\times$, and $5\times$. Consistent with the backbone requirements, these multi-scale patches were resized to 1024 (for CONCH) or 512 (for UNI) prior to feature encoding.

**Evaluation Protocol.** To ensure the statistical robustness and reliability of our results, we implemented a **5-fold cross-validation** strategy for all experiments. The dataset was randomly partitioned into five subsets at the patient level, ensuring no data leakage between training and testing splits. The final reported metrics represent the average performance aggregated across these five folds.

*Table 5.* Detailed parameters of the model architecture and their corresponding values.

| Model Architecture Parameters | Value |
|---|---|
| Number of internal ticks | 80 |
| Number of internal ticks per scale | 20 |
| Dimension of the model | 4096 |
| Dropout rate | 0.05 |
| Dimension of the input | 1024 |
| Number of attention heads | 16 |
| Synapse depth | 12 |
| Number of neurons to use for output synch | 150 |
| Number of neurons to use for action synch | 150 |
| Protocol for selecting neuron subset | Random |
| Number of neurons paired self-to-self | 0 |
| Length of the pre-activation history | 30 |
| Use deep memory | TRUE |
| Hidden dimensions of the memory | 64 |
| The number of patches selected per layer | 10 |
| Confidence threshold | 0.9 |
| Number of scales | 4 |
| The length of the synchronous output history | 80 |
| The length of the attention list | 80 |

*Table 6.* Detailed training parameters and their corresponding values.

| Training Parameters | Value |
|---|---|
| Batch size | 1 |
| Epochs | 50 |
| Learning rate | 5e-5 |
| Warmup steps | 5000 |
| Use a learning rate scheduler | TRUE |
| Type of learning rate scheduler | cosine |
| Learning rate scheduler intervals | 8000 |
| Learning rate scheduler gamma for multistep | 0.1 |
| Weight decay factor | 0 |
| Gradient quantile clipping value | -1 |
| Num workers training | 1 |
| AMP autocast | FALSE |

## D. Detailed Configurations of Slide-level Foundation Models

In addition to standard MIL approaches, we extended our evaluation to include recent large-scale slide-level foundation models: CHIEF, GigaPath, PRISM, and TITAN. These models currently define the state-of-the-art in WSI analysis but often incur substantial computational overheads.

To strictly benchmark PathCTM against these powerful baselines, we conducted experiments using their official pretrained weights and recommended preprocessing pipelines. Specifically, we aligned the feature extraction process with the original protocols of each model (e.g., using Virchow for PRISM and CTransPath for CHIEF) to ensure their best possible performance. The specific hyper-parameter settings and architectural choices for these models are listed in Table 7. This rigorous setup allows for a direct assessment of whether PathCTM can achieve comparable or superior diagnostic accuracy with significantly improved inference efficiency.

*Table 7.* Detailed configuration of the foundation models.

| Model | Patch Encoder | Patch Size | Magnification | Embedding Dim. |
|---|---|---|---|---|
| CHIEF | Ctranspath | 256 | 10× | 768 |
| GigaPath | GigaPath | 256 | 20× | 1536 |
| PRISM | Virchow | 224 | 20× | 2560 |
| TITAN | CONCH V1.5 | 512 | 20× | 768 |

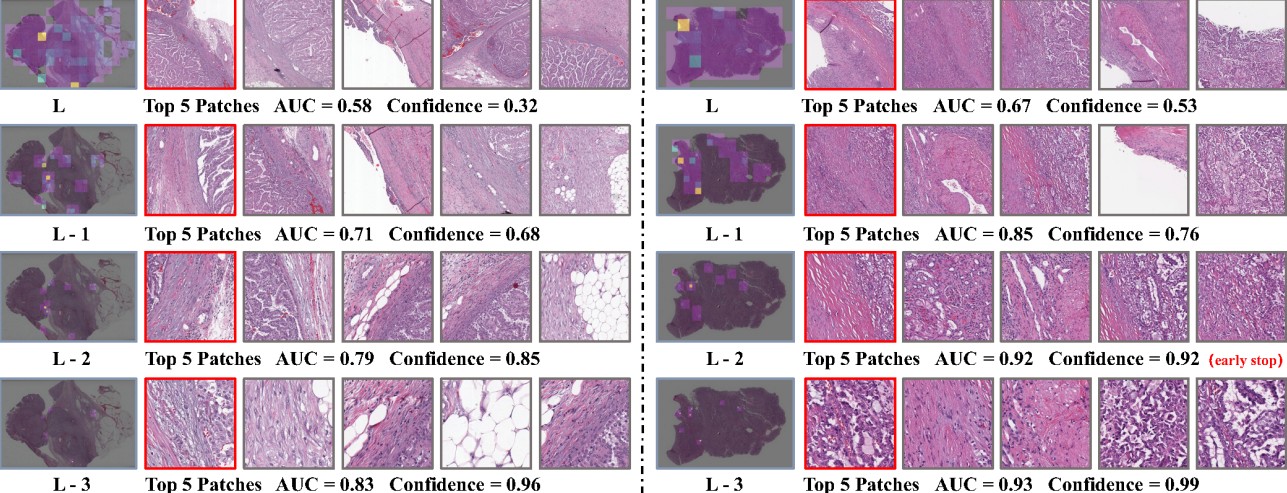

*Figure 6.* Visualization of instance-level adaptive reasoning on the RCC staging task. The figure illustrates inference trajectories for two distinct cases within the same diagnostic task. The top-5 attended patches at each scale consistently focus on tumor-normal interface infiltration, a key pathological feature for staging. (Left) A challenging case where the model accumulates evidence progressively, requiring deep reasoning up to Scale L-3 to reach sufficient confidence (0.96). (Right) A straightforward case where the model achieves reliable confidence ($\delta \geq 0.9$) at Scale L-2, triggering early stopping. This demonstrates PathCTM's capability to dynamically adjust inference depth based on the specific difficulty of individual instances.

## E. Additional Interpretability Analysis on RCC Dataset

PathCTM demonstrates robust adaptive capabilities across both different diagnostic tasks and varying case difficulties. While Figure 3 in the main text illustrates the model's adaptability across BRACS tasks, Figure 6 extends this analysis to the RCC staging task, visualizing how PathCTM dynamically adjusts its reasoning depth in response to instance-level difficulty.

Morphologically, the model performs a progressive cross-scale reasoning process. As it accesses richer semantic information at higher magnifications, both confidence and classification probability consistently improve. Notably, the top-5 high-confidence patches at each scale are predominantly distributed along tumor margins. These regions effectively capture tumor-normal interface infiltration, a diagnostically critical feature for accurate staging.

Behaviorally, the two cases exhibit distinct levels of diagnostic complexity. With the confidence threshold set to 0.9, the

harder case (left) requires deep reasoning up to the L-3 scale to achieve a reliable prediction. In contrast, the easier case (right) allows for early termination at the L-2 scale without progressing to higher magnifications. These results further corroborate that PathCTM can adaptively allocate computational resources based on the specific complexity of each case, striking an optimal balance between flexibility and inference efficiency.

## F. Detailed Parameter Discussion

We conducted a systematic analysis on four datasets using CONCH (V1.5) as the feature extractor to examine how different values of $K$ influence the effect of the confidence threshold $\delta$ on prediction accuracy and inference efficiency. The results for $K = 5$ are presented in Table 8, those for $K = 15$ are shown in Table 9, and the results for $K = 20$ are reported in Table 10.

We observed that, except for the RCC staging task—where prediction accuracy shows slight fluctuations when the confidence threshold approaches 1.0—all other tasks exhibit a consistent trend across different values of $K$: prediction accuracy tends to stabilize when $\delta$ lies within the range of 0.8 to 1.0. This finding is consistent with the results obtained for $K = 10$. Overall, the choice of the confidence threshold is largely unaffected by variations in $K$.

It is worth noting that, under the same confidence threshold, prediction accuracy does not increase linearly as $K$ becomes larger, and may even decline in certain cases. This is because, once the model has gathered sufficient patch information, providing additional patches can introduce irrelevant or redundant signals, thereby affecting the accuracy of the diagnosis. These results demonstrate that, in WSI analysis, more patches are not necessarily more beneficial.

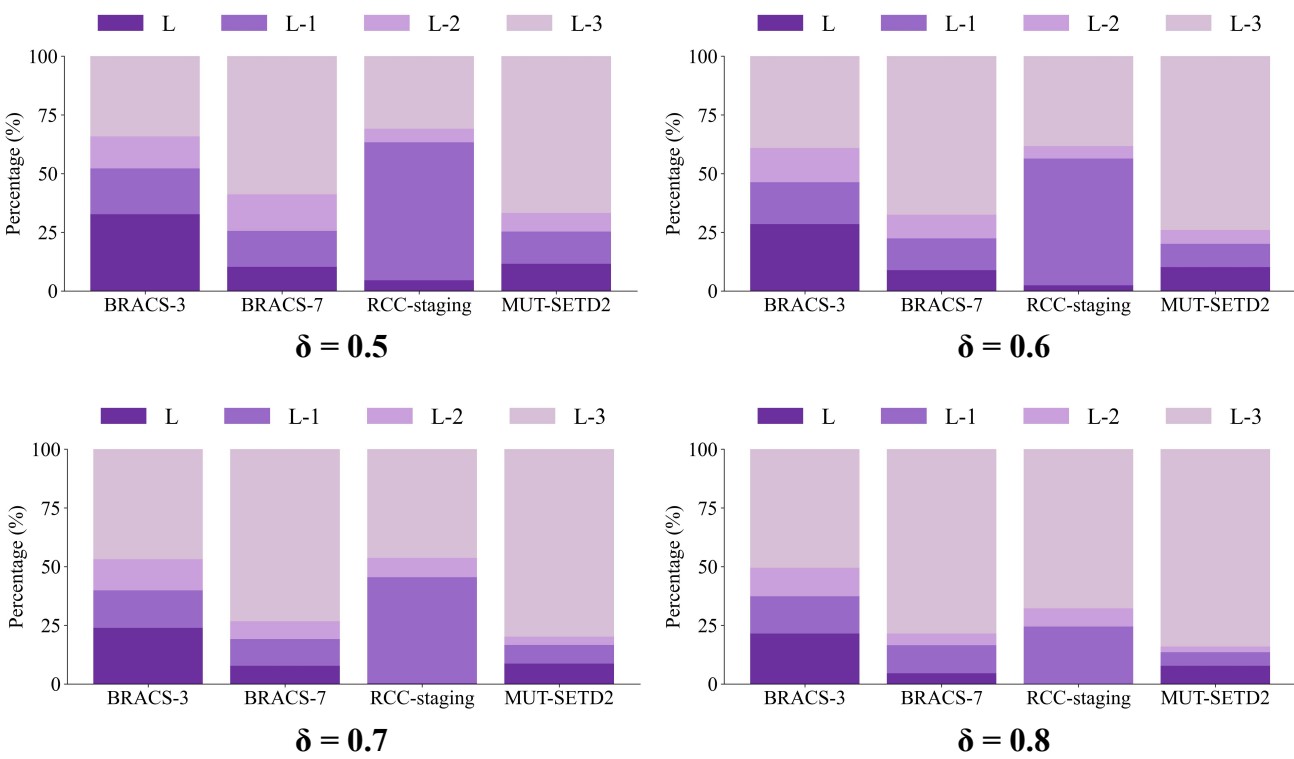

*Figure 7.* The distribution of early-stopping scales across four datasets under varying confidence thresholds $\delta \in \{0.5, 0.6, 0.7, 0.8\}$. The experiments were conducted using the CONCH feature extractor. The stacked bars illustrate the percentage of samples terminating at each magnification scale (from coarse L to fine L-3). As the threshold $\delta$ increases, the model adaptively shifts towards deeper reasoning, resulting in a higher proportion of samples proceeding to finer scales (L-2 and L-3) to satisfy the stricter certainty requirement.

To further investigate the behavioral dynamics controlled by the confidence threshold, Figure 7 visualizes the distribution of early-stopping scales across varying $\delta \in \{0.5, 0.6, 0.7, 0.8\}$. A clear monotonic trend is observed: as $\delta$ increases, the proportion of samples terminating at the coarse scale (L) significantly decreases, while the fraction advancing to the finest scale (L-3) steadily rises. This confirms that $\delta$ effectively regulates the depth of reasoning—forcing the model to accumulate fine-grained evidence when higher certainty is demanded. Furthermore, the distribution varies across tasks; for instance, the challenging MUT-SETD2 task consistently requires deeper reasoning frequencies than the simpler BRACS-3 task,

*Table 8.* The impact of the confidence threshold $\delta$ on prediction accuracy and inference efficiency when $K = 5$.

| Dataset | Metric | 0.1 | 0.2 | 0.3 | 0.4 | 0.5 | 0.6 | 0.7 | 0.8 | 0.9 | 1.0 |
|---|---|---|---|---|---|---|---|---|---|---|---|
| BRACS-3 | AUC | 89.58 | 90.06 | 90.91 | 91.48 | 92.4 | 92.69 | 92.81 | 92.99 | 93.12 | 93.24 |
| | No. Pat. | 94 | 102 | 108 | 114 | 119 | 124 | 128 | 132 | 137 | 158 |
| BRACS-7 | AUC | 87.71 | 87.69 | 88.3 | 88.54 | 89 | 89.27 | 89.39 | 89.37 | 89.38 | 89.42 |
| | No. Pat. | 83 | 104 | 114 | 122 | 129 | 134 | 140 | 144 | 147 | 158 |
| RCC-staging | AUC | 63.65 | 67.68 | 68.04 | 72.96 | 77.48 | 77.78 | 79.64 | 79.05 | 79.21 | 82.36 |
| | No. Pat. | 80 | 88 | 93 | 100 | 105 | 111 | 116 | 123 | 132 | 148 |
| MUT-SETD2 | AUC | 50.61 | 58.27 | 66.59 | 70.2 | 71.6 | 72.62 | 73.07 | 72.9 | 73.03 | 73.03 |
| | No. Pat. | 76 | 94 | 105 | 115 | 125 | 135 | 144 | 150 | 154 | 158 |

*Table 9.* The impact of the confidence threshold $\delta$ on prediction accuracy and inference efficiency when $K = 15$.

| Dataset | Metric | 0.1 | 0.2 | 0.3 | 0.4 | 0.5 | 0.6 | 0.7 | 0.8 | 0.9 | 1.0 |
|---|---|---|---|---|---|---|---|---|---|---|---|
| BRACS-3 | AUC | 88.88 | 88.65 | 89.07 | 90.28 | 91.48 | 91.59 | 92.35 | 92.81 | 93.23 | 93.91 |
| | No. Pat. | 94 | 116 | 131 | 142 | 155 | 168 | 176 | 188 | 209 | 327 |
| BRACS-7 | AUC | 86.22 | 86.78 | 87.61 | 88.04 | 88.27 | 88.54 | 88.82 | 88.99 | 89.23 | 89.42 |
| | No. Pat. | 112 | 153 | 185 | 209 | 234 | 251 | 266 | 280 | 293 | 333 |
| RCC-staging | AUC | 69.03 | 74.04 | 72.45 | 74.31 | 78.43 | 79.51 | 80.76 | 80.41 | 80.6 | 81.26 |
| | No. Pat. | 115 | 138 | 162 | 177 | 200 | 212 | 229 | 248 | 282 | 342 |
| MUT-SETD2 | AUC | 48.21 | 58.02 | 70.5 | 71.68 | 72.31 | 72.92 | 73.05 | 73.02 | 72.98 | 73.00 |
| | No. Pat. | 107 | 156 | 197 | 244 | 266 | 302 | 314 | 325 | 337 | 344 |

*Table 10.* The impact of the confidence threshold $\delta$ on prediction accuracy and inference efficiency when $K = 20$.

| Dataset | Metric | 0.1 | 0.2 | 0.3 | 0.4 | 0.5 | 0.6 | 0.7 | 0.8 | 0.9 | 1.0 |
|---|---|---|---|---|---|---|---|---|---|---|---|
| BRACS-3 | AUC | 89.13 | 89.67 | 90.02 | 90.59 | 91.01 | 92.05 | 92.4 | 92.81 | 93.31 | 93.62 |
| | No. Pat. | 121 | 157 | 182 | 203 | 220 | 240 | 259 | 280 | 309 | 410 |
| BRACS-7 | AUC | 86.11 | 86.52 | 86.53 | 86.9 | 87.01 | 87.48 | 87.77 | 88.1 | 88.43 | 88.79 |
| | No. Pat. | 118 | 156 | 186 | 218 | 248 | 276 | 303 | 320 | 342 | 413 |
| RCC-staging | AUC | 73.65 | 79.09 | 78.71 | 77.08 | 78.87 | 74.84 | 76.4 | 75.63 | 76.54 | 81.03 |
| | No. Pat. | 140 | 185 | 221 | 244 | 269 | 290 | 311 | 331 | 357 | 420 |
| MUT-SETD2 | AUC | 57.05 | 63.28 | 66.97 | 70.83 | 72.2 | 72.42 | 72.77 | 72.8 | 72.8 | 72.85 |
| | No. Pat. | 156 | 213 | 256 | 292 | 325 | 352 | 376 | 393 | 404 | 423 |

demonstrating PathCTM's ability to adaptively align its computational effort with both instance-level uncertainty and task-level complexity.

## G. Limitations and Future Work

Computational efficiency has become a key consideration in recent model development (Bae et al., 2023; Ke et al., 2025). While PathCTM significantly optimizes computational efficiency through the "thinking-in-scales" paradigm, certain limitations remain regarding clinical deployment. The hard attention mechanism introduces an inherent information bottleneck: diagnostic cues lacking salience at low resolutions—such as micrometastases or isolated tumor cells—may be irreversibly pruned, resulting in the loss of valid information. Although our framework effectively blocks the propagation of errors (false positives) to higher scales via joint attention calibration, it currently lacks a mechanism to recall valid regions that were erroneously discarded at earlier stages. This limitation inevitably restricts the completeness of effective diagnostic

information. In future work, we plan to introduce a **dynamic recall mechanism** to address this constraint. Additionally, the current inference process employs a fixed number of reasoning steps at each scale; we intend to explore strategies for **adaptively allocating the computational budget** based on information richness to further enhance efficiency and fully exploit potential information.

## H. Detailed Clarification of Runtime Measurement

In this paper, "Patch Tiling" is defined as the complete data preparation pipeline from the raw WSI to the stage before patches are fed into the encoder. This process includes not only the generation of patch coordinates, but also computationally expensive sliding-window decoding, multi-scale coordinate mapping, and I/O-intensive image patch extraction. In frameworks such as CLAM, these I/O costs are often hidden within the feature extraction loop. In contrast, this paper explicitly separates I/O operations from model inference, thereby more accurately characterizing the true computational bottleneck of the end-to-end analysis pipeline, as shown in Figure 8.

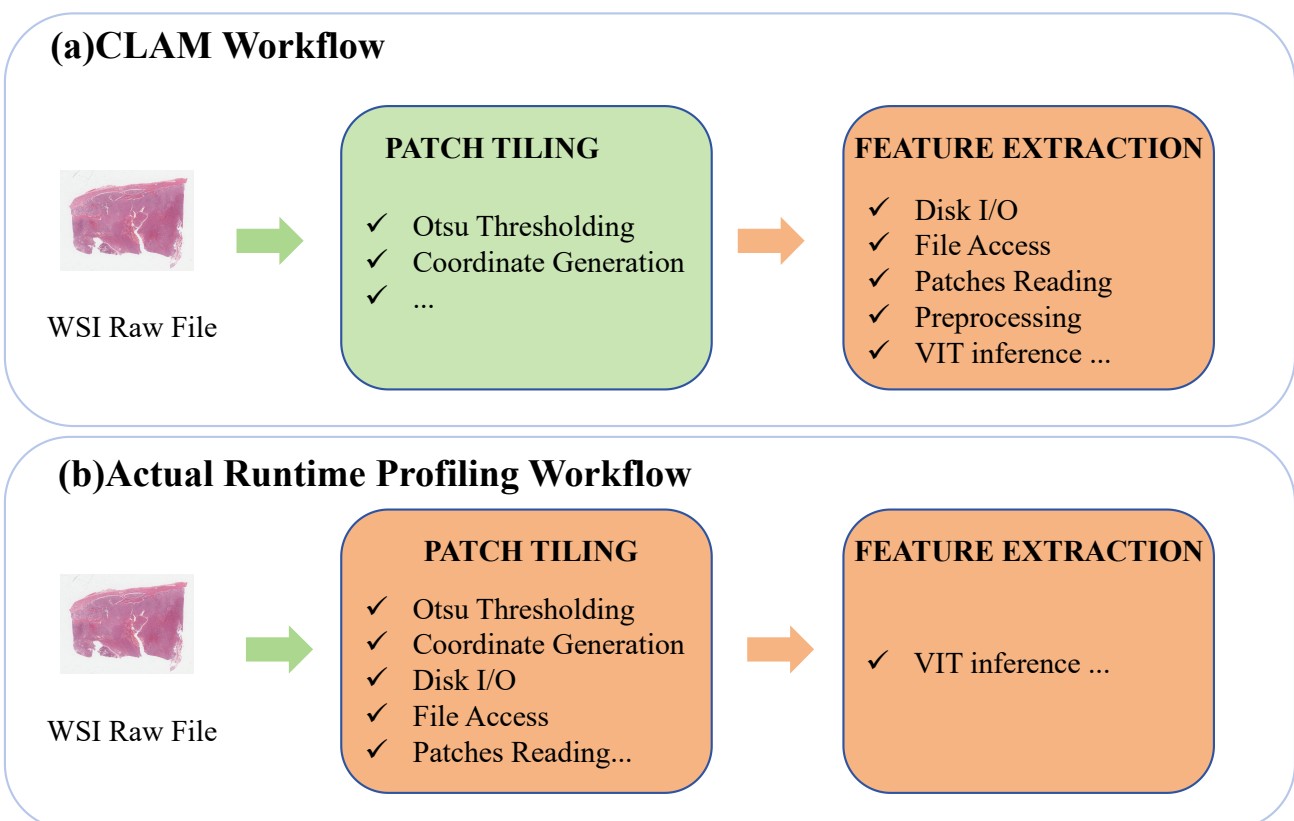

*Figure 8.* Comparison between the CLAM-style workflow and actual runtime profiling workflow.

