# OpenReview forum: "Thinking in Scales: Accelerating Gigapixel Pathology Image Analysis via  Adaptive Continuous Reasoning"
_ICML.cc/2026/Conference — ICML 2026 regular_

### Official Review · Reviewer_QMzk · 2026-03-02

**Soundness:** 4
**Presentation:** 3
**Significance:** 4
**Originality:** 4
**Overall Recommendation:** 6
**Confidence:** 5

**Summary:**

This paper proposes an adaptive continuous inference framework for WSI analysis, aiming to accelerate the inference process. The method reframes the diagnostic process as cross-scale dynamic sequential information tracking, integrating attention-guided region pruning and a confidence-based early stopping mechanism to achieve highly efficient WSI inference. The authors not only provide rigorous theoretical derivations for monotonic information accumulation and the early stopping mechanism, but also demonstrate across multiple datasets that the method significantly reduces inference time while maintaining or even improving predictive performance. Overall, this paper has a clear motivation, solid methodology, and demonstrates significant improvements in efficiency, performance, and interpretability over existing methods.

**Compliance With Llm Reviewing Policy:**

Affirmed.

**Final Justification:**

I would like to thank the authors for their detailed response and the efforts made to improve the manuscript. All of my concerns and questions have been addressed. The additional analyses and clarifications provided have further enhanced the quality of the paper. So I recommend accepting this paper.

**Key Questions For Authors:**

This paper proposes an innovative "Thinking in Scales" paradigm that highly aligns with the clinical intuition of pathologists and is backed by solid theoretical foundations. Experiments demonstrate that this method achieves an excellent balance between significantly accelerating inference and maintaining predictive accuracy, alongside strong interpretability. Although there are minor shortcomings regarding the details of the inference time evaluation and certain hyperparameter analyses, these do not undermine its core contributions. The overall work is solid and holds high value for clinical applications. To further improve the manuscript, it is recommended that the authors focus on addressing and clarifying the following points during the rebuttal phase: please elaborate on how actual data loading time is accounted for in the inference evaluation, discuss the potential impact of increasing the maximum cross-scale level limit, and analyze how the value of the hyperparameter K affects the setting of the early stopping threshold δ.

In summary, this is a solid piece of work with significant implications for accelerating WSI analysis.

**Limitations:**

Yes.

**Strengths And Weaknesses:**

### Strengths

1. High innovation and strong clinical relevance. Introducing the CTM to WSI analysis and extending it into a "Thinking in Scales" paradigm is a highly novel approach. This strategy of progressively transitioning from a global overview to a local fine-grained analysis, along with adaptively adjusting the inference depth based on case difficulty, naturally mimics the real-world diagnostic workflow of pathologists and effectively overcomes the limitations of existing rigid focusing strategies.

2. Comprehensive experimental evaluation. The experiments are conducted on four diagnostic tasks of varying difficulties, providing a thorough evaluation of the model's performance across diverse clinical scenarios. Furthermore, utilizing different pathological foundation models as backbones is a necessary and strong design choice, effectively validating the model's compatibility with existing foundation architectures.

3. Excellent performance and interpretability. Compared to traditional MIL methods and recent accelerated models, this approach achieves an impressive speed-up ratio while maintaining high accuracy, holding substantial value for the clinical deployment of WSI analysis. Additionally, the visual presentation of dynamic inference trajectories in Figures 3 and 6 is highly intuitive. The stepwise growth of confidence curves and target class probabilities clearly demonstrates the model's adaptive behavior when handling tasks of varying difficulties.

4. Solid theoretical foundation. The paper provides a rigorous theoretical basis for its core modules. It proves monotonic information gain via Bayesian updating, validates the rationality of attention pruning through first-order sensitivity analysis, and comprehensively discusses the relationship between the early stopping mechanism and the classification error lower bound, constructing a reliable theoretical framework for the proposed method.


### Major Weaknesses

1. The evaluation of inference time requires further clarification. The reported inference times for different models include patch tiling, feature extraction, and model inference. How was the actual data loading time accounted for in these measurements?

2. Insufficient discussion on the maximum number of cross-scale levels. The maximum number of cross-scale levels for PathCTM appears to be preset to 4. Can this value be further increased? If so, what impact would increasing this number be expected to have on model performance and efficiency?

3. Lack of analysis regarding the impact of hyperparameter K on the threshold δ. When dealing with diagnostic tasks of varying difficulties, how do different choices of K affect the optimal selection of δ? Is it necessary to adaptively select different thresholds δ for different values of K?


### Minor Weaknesses:

1. The clarity of mathematical notations and formulas needs improvement. In Section 3.2, Equation (4), the definitions of the synaptic state e^t and the pre-activation state h^t may not be immediately intuitive to readers. It is recommended to add 1-2 sentences of physical/intuitive explanation to aid comprehension.

2. Terminology should be unified. In the WSI domain, the interchangeable use of "Scale", "Resolution", and "Magnification" can be confusing to readers. It is recommended to unify these terms throughout the manuscript to eliminate potential ambiguity.

3. Some phrasing is slightly absolute or informal. It is advised to refine the wording to better fit an academic tone. For instance, the word "fails" in the sentence "Overall, existing CTM architecture fails in pathological settings." is somewhat arbitrary and could be replaced with a more objective phrase (e.g., "is sub-optimal" or "faces limitations").

---

> ### Author Rebuttal · Authors · 2026-03-30
>
> We thank the reviewer for the valuable comments on our work. Below, we provide a point-by-point response to the main concerns raised.
>
> >1.**Clarification of Runtime Measurement and Data Loading Overhead.** We thank the reviewer for this helpful reminder. The current manuscript does not explain clearly enough how runtime is measured, especially regarding the attribution of data loading time. In this work, **the reported total inference time is measured in an end-to-end manner, and thus includes the data preparation and data access overhead starting from the raw WSI, rather than only the model forward time.** More specifically, data loading time is counted in the patch tiling / patch preparation stage, which includes patch reading from the WSI, necessary decoding and preprocessing, as well as data access overhead related to coordinate indexing and cross-scale mapping; only after that do foundation model feature extraction and inference take place. Therefore, the “Time” reported in the table reflects a more deployment-realistic end-to-end evaluation cost, rather than an idealized model computation time that ignores I/O. **For clarity, we have added a visual explanation in Fig. 2 at https://anonymous.4open.science/r/R-D948/R.pdf.** In the revision, we will further clarify this runtime definition and provide a finer-grained time breakdown.
>
> >2.**Justification and Scalability of the Maximum Cross-Scale Level.** In the current experiments, the maximum cross-scale level is set to 4, but this is not a hard methodological limit. Rather, it is an empirical choice corresponding to the WSI pyramid and the multi-scale feature setting used in this study. Specifically, we use four scales, and therefore set the number of scales to 4. From the perspective of the method itself, this value can in principle be further increased, provided that these additional levels offer non-redundant discriminative information. In terms of expected impact, increasing the number of scale levels may have two opposing effects. On the one hand, **it could provide a more fine-grained progressive evidence acquisition path for difficult cases, potentially leading to modest performance gains on some tasks.** On the other hand, **more levels also introduce additional overhead in cross-scale transitions, feature management, and inference path length, while adjacent magnifications often contain substantial redundant information, so the benefit may not continue to grow.** Our analysis of other parameters also suggests that when additional information becomes redundant, accuracy gains tend to saturate while computational cost continues to increase; for example, performance improvements clearly plateau once the number of patches exceeds a certain point. Based on this trade-off, we currently use four levels as a practical balance between accuracy and efficiency.
>
>
> >3.**Relationship between K and the threshold δ.** In PathCTM, the two control different aspects of adaptive computation.
> - K determines how many high-resolution ROIs are retained at each cross-scale transition, thereby affecting how much fine-grained evidence can be acquired at each level and thus controlling the breadth of reasoning.
> - δ determines whether the model should stop further reasoning given the current evidence, and thus controls the depth of reasoning.
> Across different diagnostic tasks, changing K does not appear to substantially affect the choice of δ. As shown in **Tabs. 8–10** of the appendix, our parameter analysis indicates that, across four diagnostic tasks, even as K varies, the predictive performance of all four tasks becomes stable when δ in [0.8, 1.0]. Therefore, **it is not necessary to adaptively re-select a completely different δ for each K. That said, for particularly challenging tasks, jointly tuning K and δ may further improve the accuracy–efficiency trade-off, and we will clarify this point more explicitly in the revision.**
>
> >4.**Clarity of the mathematical notation and equations.** In Eq. 4, the current definitions of e^t and h^t are rather mathematical, and the intuitive explanation is still insufficient, which indeed increases the reading burden. In the revision, we will **add corresponding intuitive explanations to help readers better connect the equations with the model’s dynamic reasoning process.**
>
> >5.**Terminology standardization and clarification of definitions.** In the current manuscript, the use of terminology may cause ambiguity. In the revision, we will **standardize the terminology and provide explicit definitions at first occurrence to improve overall consistency and readability.**
>
> >6.**Improving wording for academic precision.** Some expressions in the current manuscript are slightly too absolute. In the revision, we will **perform a systematic language polishing throughout the paper so that the framing of the contributions better matches the strength of the supporting evidence.**

---

> > ### Author Rebuttal · Reviewer_QMzk · 2026-04-03
> >
> > I would like to thank the authors for their detailed response and the efforts made to improve the manuscript. All of my concerns and questions have been addressed. The additional analyses and clarifications provided have further enhanced the quality of the paper. I believe the study is now very solid and I have no further questions.

---

> > > ### Author Response · Authors · 2026-04-03
> > >
> > > We sincerely thank the reviewer for the very positive feedback and for the time and effort devoted to reading our rebuttal. We are greatly encouraged to know that our responses, additional analyses, and clarifications have adequately addressed all of the reviewer’s concerns.
> > >
> > > We also truly appreciate the reviewer’s recognition that these revisions have further strengthened the paper. We will carefully incorporate the relevant clarifications into the revised manuscript to further improve its clarity and completeness. Thank you again for your valuable feedback and support.

---

### Official Review · Reviewer_QSHG · 2026-03-13

**Soundness:** 2
**Presentation:** 3
**Significance:** 2
**Originality:** 2
**Overall Recommendation:** 3
**Confidence:** 3

**Summary:**

This paper develops a novel deep-learning approach for histopathology whole slide images (WSI). This problem space is a context where conventional computer vision approaches (e.g. end-to-end training with a ViT/CNN) struggle / fail due to the size of WSIs (e.g. up to 100,000x100,000 pixels in dimension). This paper adapts a recently developed "Continuous Thought Machine" approach to the context of whole slide images. Some theoretical results are provided to elaborate on the internal mechanisms of the model. Empirical results suggest the author's model outperforms existing standard models used in this context (e.g. CLAM) by a bit using 3 publicly available datasets. The primary impact of this paper is the reduced inference speed (e.g. 95% faster than standard models like CLAM).

**Compliance With Llm Reviewing Policy:**

Affirmed.

**Key Questions For Authors:**

Can you address the main issues raised above

a. apples-to-apples comparisons for baseline models

b. significance of inference speed

c. comparison to other fast inference speeds

d. better articulation of novelty

**Limitations:**

I do not see any potential negative societal impact.

**Strengths And Weaknesses:**

SOUNDNESS

1. MAJOR CONCERN: Did the authors perform an apples-to-apples comparison of their approach compared to baseline models? In other words, are their empirical results about accuracy valid?


The most important question here is; were hyperparameters tuned in a similar way for each model and each task? The main concern is that out-of-the-box hyperparameters were used for baseline models while more extensive hyperparameter tuning was performed for the authors models (either in a systematic or ad-hoc way). I wasn't able to find any comments addressing how the novel methods vs. baseline models were tuned in comparable ways. I'm personally open to there being more than one way to approach this, but as written I'm quite concerned the baseline models were undertuned compared to the novel models.

2. MINOR CONCERN: There are other models for WSI prediction that focus on inference speed e.g. https://arxiv.org/pdf/2502.13027. I would imagine there are simple approaches for take an already trained baseline model (e.g. CLAM) and speed up its inference speed. Examples of approaches could include: predicting attention from low resolution features, then using only the top-K patches or other model distillation approaches. Given the inference speed is a significant outcome measure in this paper I would hope to see a bit more comparison to methods that specifically target inference speed.

- 3 MINOR CONCERN: The authors do not evaluate TCGA diagnosis / outcome prediction tasks. To my understanding, the TCGA tasks are the most commonly used ones for whole slide image preidction methods like this paper. E.g. see https://arxiv.org/pdf/2206.02647 for an example of someone studying these tasks. The 3 publicly available datasets they use do appear to be valid datasets to study, but it gives me a small amount of pause they do not also study the TCGA tasks.



PRESENTATION

- 4 MOSTLY NEUTRAL POINT: The presentation is fine. A nice to have would be a more clear articulation of the few main ideas underlying the novel method (e.g. if you squint your eyes, ware are say the 3 key aspects of this model that make it work and are different from existing models).


SIGNIFICANCE

There are two key performance measures in this context 1) model accuracy and 2) model inference speed.

- 5 MODERATE GOOD THING: The empirical results demonstrating accuracy improvements in Tables 1 and 2 show modest performance improvements over a number of baseline models. The authors explore 7 different architectures and 2 foundational models in Table 1 (and 3 architectures in Table 2). They find consistent couple point AUC improvements. While these performance improvements are moderate, I suspect they are in-line with standards for publication in venues like ICML. I also believe the number baseline models / foundational models is sufficient. Deeper experiments would be appreciated, but I don't think strictly necessary for a conference like ICML.

Major concern 1 discussed above about apples-to-apples experiments gives me significant pause here. If one is not worried about this major concern (e.g. it's addressed or you as a reviewer don't find this to be a major concern) then I suspect this accuracy improvement is sufficient to warrant at least a weak acceptance.

- 6. MAJOR CONCERN: The most dramatic result is the improvement of inference speed, which raises the question: how important is it to optimize inference speed for whole slide image prediction? I can image reasonable people having different perspectives on this question. My opinion is optimizing inference speed a minor goal, but not a major goal. Stated differently, if you could make inference speed instantaneous I believe it would have a modest impact on the field, but not a major impact.

For what it's worth I've worked in computational pathology for ~5 years.  I've certainly seen other paper submissions target inference speed, but I've never seen a convincing argument that inference speed is a major goal. I absolutely may be missing something here. At the very least the authors should make a convincing argument that inference speed is of major importance (or qualify in their presentation that inference speed is of minor importance).

I can see inference speed being valuable for two reasons: 1) arriving more quickly at a diagnosis and 2) mitigating computational costs (e.g. cloud GPU spend). The first point (faster diagnosis) is not generally a major concern since models take O(seconds)-O(minutes) to run while it generally takes 1-2 days to produce a physical slide (and probably O(minutes) to O(hours) to scan). Saving a few minutes from something that takes hours does not seem like a major impact. I can think of at least two specific circumstances where time to prediction could be clinically relevant i) frozen-section assessment and ii) rapid on-site evaluation (ROSE) https://www.thelancet.com/journals/ebiom/article/PIIS2352-3964(22)00206-7/fulltext. My understanding is that frozen sections take ~20 minutes to process so a couple of minutes could be a bit of a headache, but you would have to ask a Pathologist or surgeon how bit of a deal a couple of extra minutes really is. My naive guess is that for ROSE a few minutes additional would be annoying but not a deal breaker. Please note AI is not currently deployed at scale in clinical settings (that I'm aware of) and frozen section / ROSE applications are probably a small minority of slides.


ORIGINALITY


- 7. The main source of originally is adapting the continuous time reasoning model to pathology whole slide images. My best guess is this adaptation counts for a minor to moderate amount of novelty, but I'm not sure. I'm open to it being significantly original. I'm not familiar enough with the work it's based on to assess how difficult it was to adapt to whole slide images or how ingenious their solutions were to the micro problems they faced.

OTHER


p2 "They dynamically integrate cross-scale context and halt inspection once sufficient diagnostic evidence is gathered, thereby ensuring both efficiency and reliability"

- How often do pathologists actually do this? My naive guess is that sometimes they examine the tissue this way and sometimes they examine the tissue more exhaustively. If you put a gun to my head I would guess this statement reflects how pathologists examine say 50% of cases. I am not a pathologist (I've been working in AI an pathology for ~5 years so I'd put a small amount of credibility in my prior)


p3 "composite objective that jointly rewards prediction accuracy and certainty"

- is this a good thing?


prop 1 seems like a routine result that follows from construction. I may be missing something here (e.g. perhaps there is some non-trivial insight that this proposition provides or took to identify this is an interesting proposition). If there is not some sufficiently interesting insight then formally proving this proposition feels like adding proof a proof for the sake of adding a proof.

---

> ### Author Rebuttal · Authors · 2026-03-30
>
> We thank the reviewer for the valuable comments on our work.
>
> >1.**Fairness of the baseline models.** We understand the reviewer’s concern about whether the comparisons with the baselines were conducted fairly. To address this, we have **added more detailed experimental settings for the baseline models.** Specifically, for each task, we conducted an extensive grid search over the baseline hyperparameters on the validation set, rather than simply using default settings. All models were evaluated under the same 5-fold cross-validation protocol, with strict patient-level data splits to prevent information leakage. **To improve transparency and reproducibility, we have listed the search ranges of all hyperparameters in the table below, and further provided the exact search space and final selected hyperparameters for each model in Tab. 3 at https://anonymous.4open.science/r/R-D948/R.pdf.** We hope these additions make it clearer that the reported accuracy results were obtained under comparable and fair tuning settings.
> |Optimizer|LR|WD|Dropout|Retention rate r|Chebyshev polynomial orders K|
> |:-:|:-:|:-:|:-:|:-:|:-:|
> |Adam|{1e-5,5e-5,1e-4,2e-4}|{1e-5,1e-4}|{0.1,0.2,0.3}|[0.5,1]|{4,8,12,16}|
>
> >2.**Significance of inference speed.** The efficiency gains emphasized are not only about shortening per-case processing time, but more importantly about reducing the compute and storage burden of hospital-scale screening and deployment, while improving overall throughput and cost controllability. Studies such as HDMIL, SMT, and EAGLE have also discussed the significance of accelerating WSI inference. In real-world settings, slides do not usually arrive one by one; instead, they are prepared, scanned, and ingested into digital pathology systems in batches. Large medical centers may generate thousands of slides per day. If exhaustive inference is still applied, computational cost, storage demand, I/O overhead, and cumulative processing time will all grow rapidly, becoming a practical bottleneck for foundation model deployment.
>
> >3.**Comparison to other fast inference methods.** We agree that adding comparisons with acceleration-oriented methods would strengthen the persuasiveness. We would also like to clarify that HDMIL (CVPR 2025) in Table 1 of the original manuscript is itself a recent method designed to accelerate WSI inference. To provide a more comprehensive comparison, we further **added four baselines that also aim to improve inference efficiency. Their average results across the 8 task settings are summarized in the table below, and detailed results can be found in Table 2 at https://anonymous.4open.science/r/R-D948/R.pdf.** PathCTM still maintains a more competitive accuracy–efficiency trade-off.
> |Metric|HAG-MIL|ZoomMIL|EAGLE|CLAM-Pruning|PathCTM|
> |-|:-:|:-:|:-:|:-:|:-:|
> |AUC|77.29|81.99|77.86|80.24|85.00|
> |NO. PAT.|1255| 726|1081|1070|300|
> |TIME|31.90|18.07|27.94|27.79|7.61|
>
> >4.**Better articulation of novelty.** Because CTM is a very new concept, it can easily lead to misunderstandings among readers. Adapting standard CTM—originally developed for single-scale, low-resolution natural images—to gigapixel WSIs faces severe challenges, including scale-dependent information loss and the prohibitive cost of high-resolution computation. **PathCTM introduces fundamental architectural innovations to address these difficulties and resolves three key issues:**
> - It proposes “Thinking in Scales,” transforming continuous reasoning from single-scale temporal iteration into sequential evidence search across magnifications.
> - It introduces a state-continuous cross-scale reasoning and fusion mechanism, breaking away from the discrete and one-way procedural transitions used in traditional methods.
> - It combines attention pruning with confidence-aware early stopping, moving beyond rigid patch-selection pipelines toward a conditional computation framework that decides where to look, how deeply to inspect, and when to stop, thereby achieving a better accuracy–efficiency trade-off.
>
> >5.**Additional experiments on TCGA datasets.** We agree that performance on TCGA is important for validating the generalizability of the model. We would like to clarify that, as stated in Sec. 4.1, the RCC-staging dataset used in this work is in fact derived from the TCGA project. **To further address this concern, we additionally included experiments on the TCGA-LUNG dataset. Detailed results are provided in Table 2 at https://anonymous.4open.science/r/R-D948/R.pdf.**
>
> >6.**Others.**
> - Pathologists’ slide-reading behavior is task-dependent, and not all cases follow the same search strategy.
> - The joint objective is intended to make the training objective more consistent with the confidence-based stopping criterion at inference time.
> - Prop. 1 is intended to provide a theoretical perspective on progressive cross-scale refinement, and serves as design motivation for PathCTM. We will soften this part of the presentation to avoid over-interpretation.

---

> > ### Author Rebuttal · Reviewer_QSHG · 2026-04-03
> >
> > 5. Thank you for pointing out that your RCC staging is from TCGA I did miss the detail. I should have been more precise with my comment; it would be good to see TCGA SUBTYPING task results that are fairly standard (e.g. https://arxiv.org/pdf/2206.02647 Table 1) so that we can get a rough sense of how your results compare to other results in the literature.  I do not see the LUAD results in Table 2 of the attached link.
> >
> > 1. I'm still concerned about the comparisons to baseline models.
> >
> > 1a. From reading the "Evaluation Protocol" on p14 + the above language it seems that hyperparameter tuning was performed on the test set. Hyperparameter tuning request 3 separate sets i.e. train + val + test. From reading the above I can only make out the paper uses a train + test set (and of course repeates this 5 times over mutually exclusive test folds). E.g. there is no mention that I can find of a train / validation proportion. It does seem that the authors performed the same hyperparameter tuning protocol (good or bad) on each model.
> >
> > 1b. In table 4 (I think you meant 4 not 3 in your above comment) I only see one set of selected hyperaparmeters per model task; shouldnt there be a selected hyperparameter per model / task combination?
> >
> > 2. Significance of inference speed. I agree that other machine learning papers have studied this problem, but that does not make this problem significant. Upon initial thought inference speed does *sound* like it would be significant (e.g. "big images - surely inference speed must be an issue"). A bit more though leaves the picture more cloudy. I can see three reasons inference speed would matter: 1) time sensitivity to getting results (e.g. for frozen sections you probably don't want it to take more than a few minutes of inference while the patient is under anesthesia)  2) financial cost of computation (e.g. cloud spend on GPUs) and 3) total wall clock time (assuming one does not have access to parallelization e.g. on cloud). When I've looked at ballpark estimates of time + cost for a typical pathology lab none of these appeared significant. E.g. the marginal cloud GPU cost for analysing one WSI with CLAM is fractions of a penny. It normally takes 24-48 hours for the slides (e.g. using FFPE not frozen) to actually be processed after the tissue is taken out of the patient. Waiting a few minutes does not seem like a big deal here.
> >
> > I am very willing to be convinced that inference speed is a significant problem in computational pathology. Concrete evidence (e.g. a ballpark quantitative argument) needs to be provided to support this claim.

---

> > > ### Author Response · Authors · 2026-04-04
> > >
> > > > **1. Regarding the LUAD results and the newly added TCGA RCC Subtyping experiment.**
> > > - We sincerely apologize for the typo in our previous response, which may have caused confusion during your review. The LUAD results mentioned earlier are actually reported in Table 5, rather than Table 2.
> > >
> > > - **Added TCGA-RCC Subtyping Experiments**: We agree that utilizing the standard TCGA subtyping task better demonstrates our model's performance relative to existing literature. Accordingly, referencing the paper you provided (https://arxiv.org/pdf/2206.02647 Table 1), we have included additional comparative experiments for the TCGA-RCC subtyping task. For your convenience, we have consolidated and updated the complete results for both the LUAD and the new TCGA-RCC subtyping tasks in Table 5 of the following link: https://anonymous.4open.science/r/R-D948/R.pdf.
> > >
> > > We hope these additional benchmark results provide a clearer validation of our conclusions. Thank you again for your valuable suggestions, which have helped improve the rigor and completeness of our work.
> > >
> > > >**2. Regarding the fairness of baseline comparisons.**  We sincerely thank you for your careful scrutiny of the details of the baseline comparisons. In response to your concern, we would like to make the following clarifications:
> > > - **Hyperparameter tuning was not performed on the test set.** During hyperparameter tuning, one fold was fixed as the test set. Among the remaining four folds, we strictly held out one fold as an independent validation set exclusively for hyperparameter search, while the other three folds were used for model training. Therefore, the actual data split was a strict **3:1:1** ratio.
> > > - In our previous response, the “final selected hyperparameters” reported in Table 4 were not chosen separately for each model–task pair. Instead, for each model, we compared validation performance across the four diagnostic tasks and selected the single hyperparameter configuration with the best average performance. The same principle was also applied to PathCTM.
> > > - All models were evaluated under the same hyperparameter search space, which ensured, to the greatest extent possible, the fairness of the baseline comparisons.
> > >
> > > > **3. Significance of inference speed.** We understand your logic regarding the calculation: from the perspective of a conventional FFPE lifecycle for a "single slide," saving a few minutes of inference time might indeed appear marginal.
> > >
> > > However, we wish to clarify that slide preparation and WSI inference are two entirely separate stages in the pathological workflow, governed by different bottlenecks and resource constraints. The former is tied to wet-lab processing and batch laboratory protocols, whereas the latter directly dictates the analysis efficiency once slides enter the digital viewing queue.
> > >
> > > According to our investigation, a large medical center processes approximately 1,000–2,000 pathology slides daily. A quantitative assessment of this daily volume reveals the following:
> > >
> > > -**Scenario: A large medical center generates ~2,000 WSIs daily.**
> > >
> > > -**Traditional Exhaustive Inference**: In the era of pathology foundation models, the cost of feature extraction has risen sharply. On average, exhaustive feature extraction and inference for a gigapixel WSI take about 3 minutes. For 2,000 slides, this equates to 6,000 minutes or 100 GPU-hours. For a workstation with a single consumer-grade GPU, a 100-hour daily workload is unacceptable. To process this volume within a workday, hospitals would need to run at least **10–20 high-end GPUs in parallel around the clock**; during peak periods, the system could become severely congested.
> > >
> > >
> > > -**Second-level Inference Acceleration**: PathCTM, for instance, reduces the computational workload by approximately 95%. The total analysis time for the same 2,000 slides drops to just 3 GPU-hours. This means a single workstation with one consumer-grade GPU can easily handle high-throughput, hospital-wide daily requirements.
> > >
> > > Furthermore, we highlight two critical factors:
> > >
> > > -**Computational Cost**: In real-world clinical settings, strict patient data privacy mandates on-site deployment of AI systems. The capital expenditure for purchasing and maintaining a local cluster of multiple enterprise-grade GPUs—along with the costs for cooling and power infrastructure—is prohibitively high for most pathology departments.
> > >
> > > -**Physician Waiting Cost in Clinical Interaction**: While upstream preparation time is a fixed reality, digital analysis latency is a directly perceived "sunk cost" for pathologists. A complex tumor case often involves 5–10 slides. If a physician must wait dozens of minutes for the AI to generate auxiliary results during review, the system becomes "clinically unusable." Response times must be compressed to seconds to truly integrate into the interactive diagnostic decision-making flow.
> > >
> > > Based on the analysis above, the acceleration of inference carries profound practical significance.

---

### Official Review · Reviewer_HDPR · 2026-03-13

**Soundness:** 3
**Presentation:** 2
**Significance:** 3
**Originality:** 2
**Overall Recommendation:** 4
**Confidence:** 4

**Summary:**

The paper directly tackles the challenge of the heavy preprocess procedures in multiple instance learning framework. The authors propose PathCTM that enables token-efficient continuous reasoning for WSI classification. Specifically, the method does not scan the whole regions of WSIs in high resolution, instead it prunes less important regions in lower resolution, with CTM and early stopping mechanism, thereby saving time consumptions. The experimental results suggests that this method generally outperform previous baselines.

**Compliance With Llm Reviewing Policy:**

Affirmed.

**Final Justification:**

I will raise my score since the authors have addressed my concerns.

**Key Questions For Authors:**

Please refer to the weakness section.

**Limitations:**

The authors have already included the limitations that I notice.

The whole design might require an error-correction mechanism, that once the model determines the initially selected regions are not the true discriminative ones, the model can rescan the WSI again to find other RoIs.

**Strengths And Weaknesses:**

Strengths:

1.	The paper is generally well-written and easy to follow. The motivation is reasonable and clear.

2.	The experimental results indicate that the proposed method generally outperform previous baselines with much less time consumption, though more discussion is warranted in the weakness section

Weaknesses:

1.	In my experience and according to previous works, the time consumption of patch tiling is usually much less than the feature extraction, because the patch tiling is based on otsu’s thresholding and feature extraction will require inference of a ViT model. The authors need to explain this in more detail in the paper.

2.	One critical concern of this method is the training/inference time trade-off. The paper is not quite clear on how the features are obtained from the foundation model. If the features are loaded on-the-fly, during training, the WSIs loading, tile patching, and feature extraction are performed repetitively, which will consume much more time than the traditional MIL pipeline, as it only precomputes once before training. Else, if the features are precomputed, then the most significant computational costs are already paid, and MIL inference time is really neglectable, as evidenced in Figure 1 (left) itself. The authors should elaborate on this very carefully.

3.	This paper proposes an attention-guided region pruning mechanism. However, it is quite relevant to HAG-MIL (IJCAI2023), named hierarchical attention-guided MIL, which utilizes the attention scores from a low resolution to guide the region selection in a higher resolution. The authors should discuss the similarities and differences between this method and properly cite this paper.

4.	The font size in Figure 1 makes this figure hard to read. Also, in Figure 2, the architecture of multi-scale continuous reasoning is unclear (right): (1) how the prediction of each layer is used, (2) the blue lines are too chaos, (3) the labels for the features are invisible, and so on.

---

> ### Author Rebuttal · Authors · 2026-03-30
>
> We thank the reviewer for the valuable comments on our work. Below, we provide a point-by-point response to the main concerns raised.
>
> >1.**Detailed clarification of runtime measurement.** We agree that Otsu thresholding itself is lightweight. However, in this study, “Patch Tiling” refers to the full data preparation pipeline **from the raw WSI to the point just before patches are fed into the encoder.** Beyond coordinate generation, this process also includes computationally heavy sliding-window decoding, multi-scale coordinate mapping, and I/O-intensive image patch extraction. In frameworks such as CLAM, these I/O costs are often hidden inside the feature extraction loop, whereas we explicitly separate I/O from model inference to more accurately expose the true end-to-end bottlenecks. **To make this clearer for the reviewer, we have added an illustration in Fig. 2 at https://anonymous.4open.science/r/R-D948/R.pdf.** **Dong et al. reached a similar conclusion in their HDMIL study.** The reviewer’s point is very valuable, and we will add a finer-grained time breakdown in the revision to avoid possible misunderstanding by readers.
>
> >2.**Clarification of Training/Evaluation Settings and the Source of Efficiency Gains.** The current manuscript does not describe the training and evaluation settings clearly enough, which may lead to misunderstanding. PathCTM adopts different feature usage strategies during training and during evaluation/deployment. During training, we use pre-extracted multi-scale features and retrieve them by index through cross-scale coordinate mapping, so the main advantage of PathCTM is not reflected in reduced training cost. The evaluation/deployment stage is different: the model no longer needs to perform exhaustive high-resolution feature extraction over the entire WSI. Instead, it first selects ROIs at low magnification, then uses coordinate mapping to extract high-resolution features only from a small number of Top-K regions for further inference. Therefore, **the core efficiency gain of PathCTM is mainly reflected in end-to-end evaluation time during deployment,** which is also more aligned with real clinical usage scenarios. **To make this clearer for the reviewer, we have also added an illustration, as shown in Fig. 3 at https://anonymous.4open.science/r/R-D948/R.pdf.** We will make this distinction explicit in the revision and revise the runtime discussion accordingly.
>
> >3.**Additional discussion of related work.** We thank the reviewer for recommending these important related studies. We have **further added four comparative baselines, including HAG-MIL. Their average results across the 8 tasks are summarized in the table below, and the detailed performance can be found in Tab. 2 at https://anonymous.4open.science/r/R-D948/R.pdf.** We will include formal citations and a more systematic discussion in the revision. Although the two methods share some similarity in the local logic of cross-scale attention guidance, PathCTM is fundamentally different at the paradigm level. HAG-MIL is essentially a discrete, one-way cascaded pipeline, whereas PathCTM reformulates WSI analysis as a unified continuous reasoning dynamical system. By maintaining cross-scale hidden-state continuity, PathCTM enables deep memory fusion between global context and local details, and further unifies ROI pruning and entropy-aware early stopping under the “Thinking in Scales” continuous reasoning paradigm. This not only improves inference efficiency, but also helps bridge the gap between machine algorithms and the coarse-to-fine continuous reasoning process used by pathologists.
> |Metric|HAG-MIL|ZoomMIL|EAGLE|CLAM-Pruning|PathCTM|
> |---------|:--------:|:----:|:------:|:------:|:--------:|
> |AUC|77.29|81.99|77.86|80.24|**85.0**|
> |NO. PAT.|1255| 726|1081|1070|**300**|
> |TIME|31.90|18.07|27.94|27.79|**7.61**|
>
> >4.**Improvements to figure readability and process presentation.** We sincerely appreciate the reviewer’s suggestions on improving figure readability. In the revision, we will enlarge the font size and key annotations in Fig. 1. For Fig. 2, we will more clearly explain the role of the prediction at each layer and its relationship to the final output, simplify the overly cluttered blue connections, and enlarge the labels for multi-scale features and intermediate variables. **We will also provide clearer process descriptions in the figure captions to further improve overall readability and interpretability.**

---

> > ### Author Rebuttal · Reviewer_HDPR · 2026-04-03
> >
> > I think the authors have addressed my concerns and I believe the authors should add these clarifications in the paper

---

> > > ### Author Response · Authors · 2026-04-03
> > >
> > > We sincerely appreciate the reviewer’s constructive suggestions and are greatly encouraged by the feedback. We also thank the reviewer for recognizing that the additional experiments, analyses, and clarifications have substantially improved the quality of our submission.
> > >
> > > We will certainly incorporate these clarifications into the revised manuscript to further improve the clarity and completeness of the paper. We hope that the revisions made in response to the reviewer’s comments may support a higher score.
> > >
> > > Thank you again for your valuable feedback and support.

---

### Official Review · Reviewer_Fzsh · 2026-03-13

**Soundness:** 2
**Presentation:** 3
**Significance:** 2
**Originality:** 3
**Overall Recommendation:** 3
**Confidence:** 3

**Summary:**

This paper proposes a pathology-oriented continuous reasoning framework for whole-slide image analysis that replaces exhaustive high-magnification processing with adaptive coarse-to-fine inference across scales.

**Compliance With Llm Reviewing Policy:**

Affirmed.

**Final Justification:**

I thank the authors for their efforts in the rebuttal. Some of my concerns have been addressed. I would like to maintain my original score.

**Key Questions For Authors:**

refer to Weaknesses.

**Limitations:**

yes

**Strengths And Weaknesses:**

Strengths

Fig 2 effectively illustrates the global-to-local trajectory and the stopping mechanism. The adaptive compute mechanism is valuable for both ML systems and medical imaging, regardless of the "continuous reasoning" framing.

Weaknesses

Information-theoretic framing feels like "math-washing." Prop 1 and Eq 5 discuss monotonic information accumulation for an ideal Bayesian observer, but don't strictly apply to the learned model. The paper should lean into its empirical strengths rather than heuristic theory.
Reliability claims for entropy-aware stopping aren't backed by evidence. I need to see ECE, Brier scores, or risk-coverage curves to believe this is a "reliability mechanism" rather than just a compute controller.
Fig 4 suggests that on harder tasks (RCC, MUT-SETD2), the model rarely stops early. The efficiency gains in Table 1 seem driven almost entirely by pruning. The narrative should reflect this.
Definitions are inconsistent. The query vector transition between Eq 10 and Eq 11 is ambiguous, and the "history variables" in Sec 3.2 are poorly linked to the implementation.
Missing discussion on prior work in adaptive ROI selection and multi-scale encoding. The individual components aren't entirely novel; the paper needs better positioning to justify its novelty claims. (e.g. Differentiable Zooming for Multiple Instance Learning on Whole-Slide Images, PathReasoning, Robust ROI Detection in Whole Slide Images Guided by Pathologists etc.)
A 2.3% AUC gain is modest, and the cluttered formatting in Table 1 makes it hard to parse mean/std. This undermines the empirical argument.
Table 2 only uses BRACS and inconsistent encoders. This isn't enough to claim superiority over SOTA slide-level foundation models.
Discarding subtle diagnostic cues at low-res is a major clinical risk. This shouldn't be a footnote—it needs to be quantified, especially for rare lesions or mutation prediction.

---

> ### Author Rebuttal · Authors · 2026-03-30
>
> We thank the reviewer for the valuable comments on our work. Below, we provide a point-by-point response to the main concerns raised.
>
> >1.**Theoretical interpretation of Prop. 1 and Eq. 6.** We agree that the current presentation of Prop. 1 and Eq. 6 is too strong. Prop. 1 and Eq. 6 are intended as a Bayesian-style analogy for the cross-scale fusion process, rather than as a strict theoretical guarantee of model behavior. In the revision, we will **explicitly soften the related claims, position them as design motivation and structural intuition, and place greater emphasis on the empirical strengths of the method.**
>
> >2.**Reliability of the early stopping mechanism.** To address this, we **added risk-coverage curves on four datasets** to directly examine whether the stopping mechanism indeed prioritizes low-risk samples for early termination. We performed the analysis under different thresholds δ∈[0.7,0.98]. The results show that as the threshold increases, the coverage of early-stopped samples gradually decreases, while their error rate consistently drops and approaches 0 at high thresholds. This indicates that early stopping is not merely a mechanism for controlling computation, but selectively terminates easier and lower-risk samples earlier under higher confidence. Detailed results are provided in **Tab. 1 and Fig. 1 at https://anonymous.4open.science/r/R-D948/R.pdf.**
>
> >3.**Efficiency gains.** The efficiency gains indeed mainly come from region pruning, while early stopping is more focused on adaptively allocating computation depth across tasks or samples of different difficulty. We will further clarify the respective roles of these two components.
>
> >4.**Notation and implementation mapping.** We will add more detailed variable explanations and clearer figure annotations to better connect the method description with the implementation details.
>
>
> >5.**Additional discussion of related work.** We thank the reviewer for recommending these important related works. We have **added four comparative baselines, and their average results across the 8 tasks are summarized in the table below; detailed performance can be found in Tab. 2 at https://anonymous.4open.science/r/R-D948/R.pdf.** We will also expand the discussion of the reviewer-recommended prior work in the revision. Compared with these methods, the core innovation of PathCTM lies in modeling WSI analysis as a continuous reasoning system under the “Thinking in Scales” paradigm. Traditional methods typically treat scale transitions as discrete, one-way procedural steps, whereas PathCTM maintains cross-scale hidden-state continuity, enabling deep memory fusion between global context and local details.
> |Metric|HAG-MIL|ZoomMIL|EAGLE|CLAM-Pruning|PathCTM|
> |-|:-:|:-:|:-:|:-:|:-:|
> |AUC|77.29|81.99|77.86|80.24|85.00|
> |NO. PAT.|1255| 726|1081|1070|300|
> |TIME|31.90|18.07|27.94|27.79|7.61|
>
> >6.**AUC gain evaluation.** To assess the AUC gains, we further conducted statistical significance analysis. The **p-values across the 8 task settings are 0.038, 0.140, 0.038, 0.583, 0.011, 0.048, 0.313, and 0.031**, respectively. Among them, 5 settings show statistically significant improvement; 2 settings still maintain overall performance gains, and 1 setting achieves the second-best result. This suggests that the experimental conclusions are not driven by random variation. Importantly, the main advantage of our method lies in accelerating inference while maintaining competitive predictive performance. Therefore, the core value of PathCTM lies in its superior accuracy–efficiency trade-off. In the revision, we will also reformat Table 1 and improve the grouping and spacing of the table.
>
> >7.**More evaluation of slide-level models and clarification of encoders.** We further **added comparative experiments on RCC and MUT (see Tab. 3 at https://anonymous.4open.science/r/R-D948/R.pdf).** Regarding patch encoders, it is common practice in computational pathology to adopt the officially recommended encoder for each foundation model, in order to avoid artificially weakening the baselines, as also followed in studies such as Virchow, TITAN, and CARE. In addition, **TITAN and PathCTM use the same encoder (CONCH), so this comparison still provides a direct and meaningful reference under matched patch representations.**
>
> >8.**Risk assessment of missing cues at low resolution.** We invited a pathologist to **conduct a quantitative review of the misclassified cases in BRACS-3, BRACS-7, and RCC-staging.** The proportions of errors caused by missing critical regions at the low-resolution stage were 2.54%, 4.45%, and 5.30%, respectively. This indicates that the risk objectively exists, but is not the primary source of error. For the MUT gene mutation prediction task, such analysis is currently not feasible because there are no directly observable morphological criteria. We will discuss this issue more explicitly in the revision.

---

> > ### Author Rebuttal · Reviewer_Fzsh · 2026-04-03
> >
> > Some of my concerns have been addressed.

---

> > > ### Author Response · Authors · 2026-04-03
> > >
> > > We sincerely appreciate the reviewer’s constructive suggestions and believe that the additional experiments, analyses, and clarifications have substantially improved the quality of our submission. We hope that these revisions further strengthen the paper and support a higher evaluation.
> > >
> > > At the same time, we also understand that the reviewer may still have further questions or concerns. We would be very grateful if the reviewer could further point out the remaining issues, and we will do our best to provide a more thorough response and clarification.

---

### Decision · Program_Chairs · 2026-04-30

**Decision:**

Accept (regular)

**Comment:**

This paper presents a pathology-oriented continuous reasoning framework for whole slide image analysis that aims to improve the accuracy and efficiency trade-off. Reviewers agree that the paper tackles an important practical problem in computational pathology and highlight potential value for real deployment. The main concerns focus on the theoretical framing being stronger than what is supported, fairness and clarity of the baseline comparison protocol, the practical significance of inference speed, and the need for better positioning against prior fast inference and adaptive multi-scale methods. The rebuttal addresses most of these concerns. While some concerns remain about presentation and the strength of a few claims, the reviewers generally view the contribution as meaningful. The authors should incorporate the clarified details in the final version.